# Financial frictions and stock return: A novel least minus more frictional factor for asset pricing models in emerging economies

Saifullah Khan[1]*, Adnan Shoaib[2], Rehan Aftab[1], Muhammad Yasir[1], Muhammad Bilal Saeed[1]

1 FAST School of Management, FAST-National University of Computer and Emerging Sciences, Islamabad, Pakistan, 2 Greater Manchester Business School, University of Greater Manchester, Islamabad, Pakistan

* saifullah.khan@nu.edu.pk

## Abstract

The primary objective of this study is to empirically evaluate the role of various levels of financial friction in explaining stock returns through different asset pricing models. This study enhances asset pricing model estimates by incorporating diverse levels of financial friction by introducing a novel least minus more frictional asset pricing factor specifically constructed for emerging economies. The empirical analysis is conducted using data from a sample including five countries: China, India, Pakistan, Bangladesh, and Sri Lanka. Monthly data from 735 listed manufacturing firms is used to estimate stock returns from 2009 to 2024. These models are rigorously tested for optimal estimation using panel data models. The findings indicated that different levels of financial friction collectively exert inverse effects on stock returns. Macro-economic and microeconomics frictions are found to be more pronounced in Pakistan compared to other countries, while financial market frictions are more acute in India, and firm-level frictions are most significant in China. The results further reveal that stock returns are overestimated in conventional asset pricing models. Incorporating different levels of financial frictions into these models substantially reduced the abnormal returns. This study has profound implications at macroeconomic, microeconomics, financial market, emerging the economies that are. Managers can leverage these insights to formulate superior strategies aimed at enhancing profitability, fostering robust business-to-business relationships, and minimizing costs across various levels. The findings enable firms to preemptively optimize their operations within the context of prevailing financial frictions.

## 1. Introduction

Since its inception in the early 1960s, the Capital Asset Pricing Model (CAPM), pioneered by Sharpe [1] has been the dominant framework for valuing risky assets in

**Data availability statement:** Complete data set is available from Harvard Dataverse open database: https://doi.org/10.7910/DVN/MGS0A3.

**Funding:** The author(s) received no specific funding for this work.

**Competing interests:** There are no competing interests to declare by the authors.

advanced economies. A lineage of scholars, including Fama and French [2], Carhart [3], Fama and French [4], and Doğan, Kevser [5], have extended and generalized the CAPM. Despite more sophisticated models, the CAPM remains essential for finance professionals due to its simplicity and broad applicability. These models are applied similarly in emerging economies [6]. However, CAPM and its extensions rely on the efficient market hypothesis and the Modigliani and Miller [7] irrelevancy principle, assuming efficient market with no taxes, transaction costs, default risk, information asymmetry, or agency costs. While these assumptions may hold in advanced economies but ap Gwilym, Ebrahim [8] argue that financial frictions are pervasive as these frictions impede market operations, contradicting traditional theory. Therefore, this study incorporates these frictions into asset pricing models to derive more realistic stock return estimates, shifting the classical paradigm from friction less to real-world conditions. Furthermore, it addresses statistical issues like weak identification and parametric instability. This study is a paradigm shift for asset pricing models, moving from efficient market assumptions to economies characterized by frictions and anomalies. This study integrates financial frictions into asset pricing models by incorporating a novel "least minus more" (LMM) factor, grounded in agency theory. It investigates whether this LMM frictional factor enhances empirical stock return estimations. This analysis contributes in three ways: First, it examines how macro-, micro-, and firm-level frictions influence and improve asset pricing model estimations. Second, it analyzes how manufacturing firm stock returns in developing countries respond to varying levels of these frictions. Third it gives the comparison between actively growing emerging economies (China and India) and stalled emerging economies like Pakistan, Bangladesh, and Sri Lanka.

Recognizing that stakeholder assessments often neglect the diverse economic constraints and financial frictions prevalent in various contexts, this study bridges this gap by examining the impact of varying levels of economic frictions on the stock returns of listed manufacturing firms in five South Asian emerging economies: China, India, Pakistan, Bangladesh, and Sri Lanka. Financial frictions, encompassing obstacles like transaction costs, information asymmetries, and regulatory constraints, impede efficient financial market functioning. These constraints hinder firms ability to secure external financing for desirable investments, potentially due to difficulties in issuing equity or debt, limited access to bank loans, credit constraints, or illiquid assets. Understanding these dynamics offers valuable insights into the complex interplay between stock returns and the economic environment [9], crucial knowledge for investors, policymakers, and firms. While financial frictions, the study of market imperfections and their effects on economic outcomes, have been extensively studied in relation to investment, growth, inequality, and monetary policy, this study extends this literature by examining how these frictions can be effectively modeled, assessed, and potentially mitigated to enhance stock return stability.

The study of financial frictions (FF), exploring market imperfections and their impact on economic outcomes, has become a prominent area within financial economics. While prior research has investigated FF's influence on investment, growth, inequality, and monetary policy [10], this study extends the literature by focusing on

how these frictions can be effectively modeled, assessed, and potentially mitigated to enhance stock return stability. Firms must comprehend the financial constraints within their operating environment to maximize profitability. Financial frictions, representing the costs and limitations firms encounter in daily operations and external financing, significantly affect firm performance, investment decisions, and financial policies. For example, short-term tax incentives can substantially boost equipment investment, particularly for small businesses. Similarly, understanding banking regulations impact on credit availability is crucial for predicting legislative effects on investment. Financially constrained firms are especially sensitive to interest rate and credit availability fluctuations, underscoring the importance of incorporating FF into investment decisions. Therefore, understanding the constraints imposed by FF is paramount for influencing investor financial and investment choices, ultimately guiding investors toward achieving and maintaining optimal stock returns and maximizing profitability.

Financial frictions are multifaceted, manifesting at various levels. Macroeconomic frictions arise from monetary, fiscal, and prudential policies [11,12]. Microeconomic (business-level) frictions stem from liquidity needs, collateral requirements, and constrained credit supply [13]. Financial market frictions are caused by information asymmetries, market inefficiencies, and contractual constraints, hindering market efficiency and liquidity [14]. Finally, firm-level frictions, such as agency costs, capital adjustment costs, and operational costs, also exist. Addressing these diverse frictions is essential for firms to optimize financial performance, growth, profitability, and survival, and for policymakers to promote a more efficient and stable financial system.

At the microeconomics level, financial frictions also impact firm interactions with financial intermediaries. Businesses often establish affiliations with these intermediaries to mitigate information asymmetry and enhance managerial oversight, though such affiliations may entail costs like elevated interest rates or fees. Understanding how financial linkages affect access to external financing enables firms to manage these trade-offs and make informed partnership decisions. In recent studies, Müller and Verner [15] highlight how financial frictions, particularly bank runs, can destabilize the banking system and constrain credit availability, disrupting these crucial interactions. Similarly, Konstantakopoulou [16] emphasizes the role of financial intermediaries in facilitating investment and growth, but notes that macroeconomic frictions like interest rates can hinder this process. Furthermore, Petersen and Rajan [17] demonstrate that financial frictions, such as information asymmetries, influence business-intermediary dynamics, especially during economic stress. These studies illustrate that financial frictions operate within business-to-business relationships and interactions with financial intermediaries. Ignoring these frictions can severely restrict firms access to capital, thereby hindering operational efficiency and success. This study addresses these financial friction issues, exploring how to achieve higher capital and cash returns, make informed financing decisions, and assess the impact of these frictions on investment behavior. Effective management of financial frictions is crucial for firm performance, as appropriate contract design can align incentives, mitigate agency problems, and unlock value. Similarly, effective cash management to mitigate financial frictions can reduce financial distress and improve performance [18].

Financial friction affects diverse economic sectors and levels. Monetary policy influences real economic activity through money and credit channels. Extensive literature acknowledges financial frictions as detrimental to various financial sectors, including housing [19], financial and capital markets [14], the corporate sector [20], and even individuals [21]. However, the comprehensive impact of these frictions on manufacturing firms within emerging economies at the micro-level remains under-explored.

Recent studies have incorporated financial frictions into New Keynesian models, addressing the zero lower bound on deposit rates and the possibility of negative reserve rates. These models highlight how macroeconomic policies, like monetary adjustments, can inadvertently create or exacerbate macroeconomic financial frictions, impairing economic efficiency, especially after negative shocks [22]. Financial intermediaries and market frictions significantly influence the monetary transmission mechanism across economies and time periods [23]. Furthermore, nonperforming loans act as a financial friction for intermediaries, leading to market inefficiency [24].

This study uniquely evaluates listed manufacturing firms in specific South Asian emerging economies, identifying the causes and exposures to financial frictions that hinder their financial development. Comparing these economies, with their varying growth levels, financial development, and economic structures, allows for a richer understanding of these frictions (informed by prior research). Financial frictions can differentially affect emerging economies (Pakistan, Bangladesh, Sri Lanka), actively growing emerging economies (China and India), and advanced economies. Since Pakistan is the main focus of the study and its comparison is made with the neighboring countries. Therefore, this study gives an interesting case of South Asian select economies. In this study two countries China and India are actively growing economies, and two countries are relatively stalled economies, i.e., Bangladesh and Sri Lanka as compared to Pakistan. Moreover, all the economies have same ecological and geographical dynamics which make comparison easier, and if we include any other country from like US, Canada, Australia, Russia or European countries then it would be difficult to make caparison with such a diverse and advanced economies. Existing literature often focuses on mixed industries (services, finance, merchandising, production), leaving a gap in understanding the unique challenges faced by manufacturing firms in emerging markets. South Asia presents a compelling case, as prior models have primarily focused on developed (US, European) or underdeveloped economies. These Asian countries possess significant growth potential, driven by catch-up growth and higher investment rates [25]. Critically, the micro-level impact of financial frictions on manufacturing firms in emerging economies, characterized by underdeveloped financial markets, weaker institutions, and information asymmetry, stays under-explored. The sample countries are dominated with the manufacturing sector as compared to the services sector and have capital- and labor-intensive industries. The manufacturing firms were selected for analysis due to its pronounced sensitivity to financial frictions, whereas the financial sector is posited as the transmission mechanism and conduit for these constraints. Given the manufacturing sector's crucial role in these economies development, understanding these impacts is vital for firm financial indicators [26]. A micro-level analysis offers granular insights into how different financial frictions affect firm value, capital structure, and stock returns [27]. Therefore, this study collects data on these firms to quantify the impact of various financial frictions on their fundamentals, developing models that incorporate the specific characteristics of emerging economies and their manufacturing sectors. It further compares these impacts across the selected economies to identify variations and best practices, ultimately contributing policy recommendations to mitigate negative effects, potentially focusing on improving financial access, strengthening institutions, and reducing information asymmetry.

Agency theory has long been employed to mitigate agency costs associated with firm cash flows [28]. Firms operate within an ecosystem of stakeholders (government, creditors, financial institutions, investors, suppliers, traders), each with vested interests in firm profitability. For example, governments collect taxes, creditors charge interest, and suppliers/traders seek profit margins. These stakeholder demands create substantial costs for firms. Conversely, firms aim to maximize profitability, creating a conflict of interest that generates agency-like costs to align stakeholder interests. Traditional agency theory applications have been limited to firm and financial market levels. This study extends the scope of theory which encompass macro- and micro-level interactions between firms (agents) and stakeholders (principals)—government, suppliers, creditors, and other firms. Theoretically, this expands agency theory to include macro- and micro-level firm interactions with government and other stakeholders, generating agency costs in the form of taxes, tariffs, trade credit, and financing costs [29]. This paper makes a significant theoretical contribution by providing a robust framework for analyzing the complex interplay between financial frictions, firm fundamentals, and macroeconomic outcomes, a framework applicable to diverse economic phenomena. Higher levels of information asymmetry (capital market friction) a key aspect of financial friction can intensify the principal-agent problem, leading to increased monitoring costs and potentially suboptimal investment decisions [30]. Conversely, reduced information asymmetry, perhaps through improved transparency mechanisms [31], might mitigate these agency costs even in the presence of other forms of financial friction, such as transaction costs. Furthermore, this study recognizes the need for empirical validation while the current draft primarily lays theoretical groundwork, we propose two avenues for addressing this in

the revised manuscript. First, we will incorporate a more thorough review of existing empirical literature that examines the interplay between financial friction and agency theory. Study by Hadlock and Pierce [32] examines financial constraints and investment provide valuable insights into how financial constraints (financial market friction) affect corporate behavior in line with agency cost arguments. This study explicitly discusses how our theoretical framework aligns with and potentially extends these empirical findings.

Second, this study outlines potential avenues for future empirical research that could directly test the relationships we propose. This involves suggesting specific datasets and econometric methodologies that could be employed to quantify the impact of different levels of financial friction on agency-related outcomes, such as executive compensation structures or corporate governance mechanisms [30].

While some studies have used "financial frictions" and "financial risks" interchangeably [33], this study distinguishes between them. Frictions represent pre-defined, measurable costs and operational hurdles, while risks are unpredictable, often ecologically driven shocks with uncertain, and immeasurable, magnitudes, negatively impacting stock returns [34]. This distinction is crucial. Frictions have known sources (e.g., taxes, regulations), predictable timing, and quantifiable magnitudes. Shocks, conversely, originate from unknown sources, occur unexpectedly, and have uncertain durations and impacts. Conceptually, frictions are predetermined costs, whereas shocks are unforeseen external events. Although short-term effects may be similar, this study employs a partial adjustment model to analyze their distinct long-term effects, empirically differentiating these concepts and measuring the evolving impact of frictions over time. This approach offers novel insights by combining internal (frictions) and external (shocks) factors, addressing a literature gap and providing a unique understanding grounded in empirical, methodological, and theoretical foundations for both heterogeneous and homogeneous listed manufacturing firms within select developing economies.

## 2. Literature review

### 2.1. Financial frictions

Macroeconomic frictions (MAF) encompass the theoretical framework for understanding and modeling macroeconomic barriers that deviate financial markets and economies from idealized equilibrium. These frictions influence market operations, economic decision-making, and overall macroeconomic performance. Macroeconomic policies, such as fiscal and monetary interventions, can either mitigate or amplify the consequences of these frictions. As highlighted by Baker et al. (2016), Economic Policy Uncertainty, EPU represents a significant macroeconomic friction that can profoundly affect economic activity. Their valuable datasets, now available for China and India, offers a compelling avenue for further investigation within our framework. Moreover, the recent work by Kundu and Paul (2022) on the impact of EPU on G-7 stock markets underscores the relevance of this factor in understanding market dynamics. Building upon this literature, future iterations of our research will aim to incorporate measures of EPU for the specific economies under examination, where data availability permits (e.g., China and potentially India). This will allow us to directly evaluate the interplay between macroeconomic policies and the identified macroeconomic frictions. Specifically, we intend to explore whether periods of heightened EPU intensify or mitigate the consequences of these frictions on stock market behavior. For example, fiscal stimulus or tight monetary policy can worsen existing frictions, potentially leading to unemployment and resource misallocation. Conversely, financial frictions, like limited credit access or high borrowing costs, can amplify macroeconomic shocks and prolong recessions. The impact of MAF is clear in the housing sector, business cycles, and the financial sector. Given the detrimental effect of deeply rooted macroeconomic frictions on corporate financial performance, this study examines their specific impact on firm stock returns. Macroeconomic shocks generate financial frictions, such as those related to monetary policy, to which firms with lower financial leverage are more sensitive due to their flatter investment financing cost curves [12]. During financial crises, financial frictions and shocks can predict macroeconomic cyclical behavior [11]. These studies provide a rationale for incorporating macro-level frictions into asset pricing models to investigate stock returns.

Macroeconomic shocks, particularly those manifested in monetary policy, differentially impact firms based on their financial leverage [12]. Firms with lower leverage, having flatter investment financing cost curves, exhibit greater sensitivity to these shocks. Financial frictions and shocks can predict macroeconomic cyclical behavior during crises [11]. Firms are vulnerable to economic shocks at various levels. These studies motivate the incorporation of macro-level frictions into asset pricing models to investigate stock returns. These frictions can limit firms access to capital, investment in growth, and liability management. For instance, Ottonello and Winberry [12] show that highly leveraged firms are more vulnerable to monetary tightening due to increased debt servicing costs, potentially impacting profitability and firm value. Conversely, firms with lower leverage and flexible financing are better positioned to withstand such shocks, influencing their stock returns and financial performance [35]. This highlights a complex interplay between macroeconomic frictions and firm performance, crucial for investors, managers, and policymakers to anticipate risks, make informed decisions, and design effective mitigation strategies. Microeconomic frictions, defined by Blyde and Pineda [36] as institutional, legal, and bureaucratic constraints increasing business costs, and by Gertler and Kiyotaki [37] as financial market imperfections like credit rationing and collateral requirements, restrict firms access to investment funds, leading to suboptimal decisions, reduced productivity, and hindered growth. Similarly, stringent labor laws increase hiring/firing costs, raising business adjustment costs, affecting firms flexibility and competitiveness. These labor market frictions also disrupt firm financial performance, a connection further explored in this study.

Microeconomic frictions (MIFs) encompass transaction costs (information gathering, contract negotiation/enforcement), which hinder mutually beneficial transactions [38]. Market power, where a few firms control a market, also creates microeconomic frictions, leading to higher prices, lower quality, and reduced innovation. Government policies, though often well-intentioned, can exacerbate these frictions through increased operating costs, price distortions, and taxes/tariffs. Bureaucratic inefficiencies, like complex licensing processes, and weak institutions that impede rights enforcement and market function, further contribute to these frictions. These frictions impede firm operations, increased risk-weighted capital requirements, increase interest rates and decrease lending/output [35]. Similarly, bank liquidity requirements reduce loan supply and output while increasing safer asset holdings. Berger and Vavra [39] offer insights into how microeconomic frictions influence consumption during recessions, suggesting that investment goods relative prices are endogenous, shock-dependent, and influenced by trade openness and investment frictions. Non-neutral shocks can increase the relative price of investment goods when investment friction decreases. Microeconomic friction constrains firm financing for growth and investment. This study uniquely examines the impact of these frictions on firm financial fundamentals (value, capital structure, stock return), specifically within emerging South Asian economies, where such research is limited.

Financial frictions are fundamentally rooted in information asymmetry, a market friction arising from information disparities between market participants [40]. This asymmetry can lead to adverse selection and moral hazard, distorting capital allocation and potentially mispricing stocks. Shen [41] recently confirmed the positive correlation between capital misallocation and financial market frictions in Chinese firms, suggesting that these frictions (information asymmetry, transaction costs, liquidity constraints) distort investment and impede capital flow to its most productive uses. Similarly, information asymmetry can depress stock prices and valuations in equity markets. This study investigates these issues in listed manufacturing firms of emerging countries, a novel area of research. Li, Liu [42] argue that the costs associated with issuing equity and the penalties for loan default contribute to financial market frictions. However, this study examines the specific channels through which these frictions affect stock returns.

Firm-level frictions arise from operational costs and constraints, manifesting as distortions and volatility in productivity, revenue, or profit, or as input costs (labor, production and operational cost) [43]. Midrigan and Xu [10] show how these frictions distort firm entry and technology adoption, leading to capital misallocation and productivity losses. Higher frictions can reduce firm profitability, growth, and stock returns due to constrained investment and suboptimal resource allocation. Financial constraints may also lead to more conservative capital structures, limiting the benefits of debt financing. While easing financial constraints can improve firm welfare, Gopinath, Kalemli-Özcan [44] show that poorly designed financial

policies can misallocate capital, highlighting the need for efficiency. Furthermore, firm-level friction-induced volatility in productivity, revenue, and profit can increase cash flow risk, potentially lowering stock returns.

This study aims to directly measure specific frictions (e.g., macro, micro, financial market, and firm level frictions) rather than relying on indirect proxies like size and market capitalization. This direct measurement can potentially capture nuances that broad proxies might miss. We acknowledge the recent extensive literature utilizing firm characteristics as proxies for financial frictions, including the important work of Ferreira, Haber [45] and Bali, Brown [46], however, our approach offers a potentially more granular and specific lens on the impact of particular frictions. Firm characteristics such as size can be influenced by several factors beyond financial constraints, making it challenging to isolate the precise effect of friction. Therefore, to counter this issue we run traditional Fama-McBeth regression on frictional factors to determine whether a proposed asset pricing model can explain the differences in average returns across a set of assets at a given point in time.

## 2.2. Stock return and asset pricing models

Stock return, the percentage change in a stocks value over a given period, is a key metric for shareholders and analysts. Positive financial outcomes and sound fundamentals typically drive higher returns [4], though other factors also play a role. Economic indicators and market conditions influence stock returns across sectors [47], as do investor sentiment and industry performance. Crucially, financial frictions significantly affect stock returns. While strong fundamentals often correlate with higher returns, these frictions can disrupt this relationship. Information asymmetry, a fundamental financial market friction, can cause adverse selection and moral hazard, leading to stock mis-pricing and distorted returns. Stock returns were initially studied by Sharpe [1], incorporating market risk and the risk-free rate. Fama and French [2] added size and value factors, and Carhart [3] included momentum. Various other factors have since been considered, with asset pricing theory, APT, agency theory, prospect theory, and rational pricing theories dominating equity return models. Stock prices, unlike other firm indicators [48], are particularly susceptible to information asymmetry. Because stock price reflects dividend distribution, derived from net income, which is reduced by frictional costs (taxes, tariffs, obligations), it is also affected by financial frictions stemming from monetary policy, regulations, and market imperfections. Brunnermeier, Farhi [49] found that greater financial frictions in the Euro area generate varying asset price and risk premium responses. Financial frictions (transaction costs, limited market participation) can hinder efficient trading, increasing volatility and distorting stock returns. Interconnected markets mean frictions in one market can spill over, affecting stock returns across sectors. As Jegadeesh and Titman [47] note, while economic conditions and investor sentiment influence returns, financial frictions can amplify these effects. During downturns, these frictions worsen, hindering firms access to capital and further depressing stock prices. Financial frictions can also influence dividend policy, with constrained firms less likely to distribute dividends. In conclusion, financial frictions are crucial determinants of stock returns, distorting prices, increasing volatility, and influencing firm decisions, all of which impact stock performance. Understanding these complex relationships is essential for investors and analysts navigating markets characterized by financial frictions.

## 3. Theoretical framework and hypothesis development

### 3.1. Theoretical framework

The underlying dominant perspective is that the financial frictions affects the asset pricing of the stock returns. The model is based on the four classical asset pricing models, i.e., Sharpe [1] capital asset pricing model (CAPM), Fama and French [2] three factor model (FF3F), Carhart [3] four factor model (Carhart 4F) and Fama and French [4] five factor model (FF5F). The relevant financial frictions are introduced in these asset pricing models to have a more robust estimate of these factors on the stock return. The theoretical framework of the study is given in Fig 1. In this figure, the least minus more frictional factor is introduced in the asset pricing models which is constructed by combining the macro-, micro-,

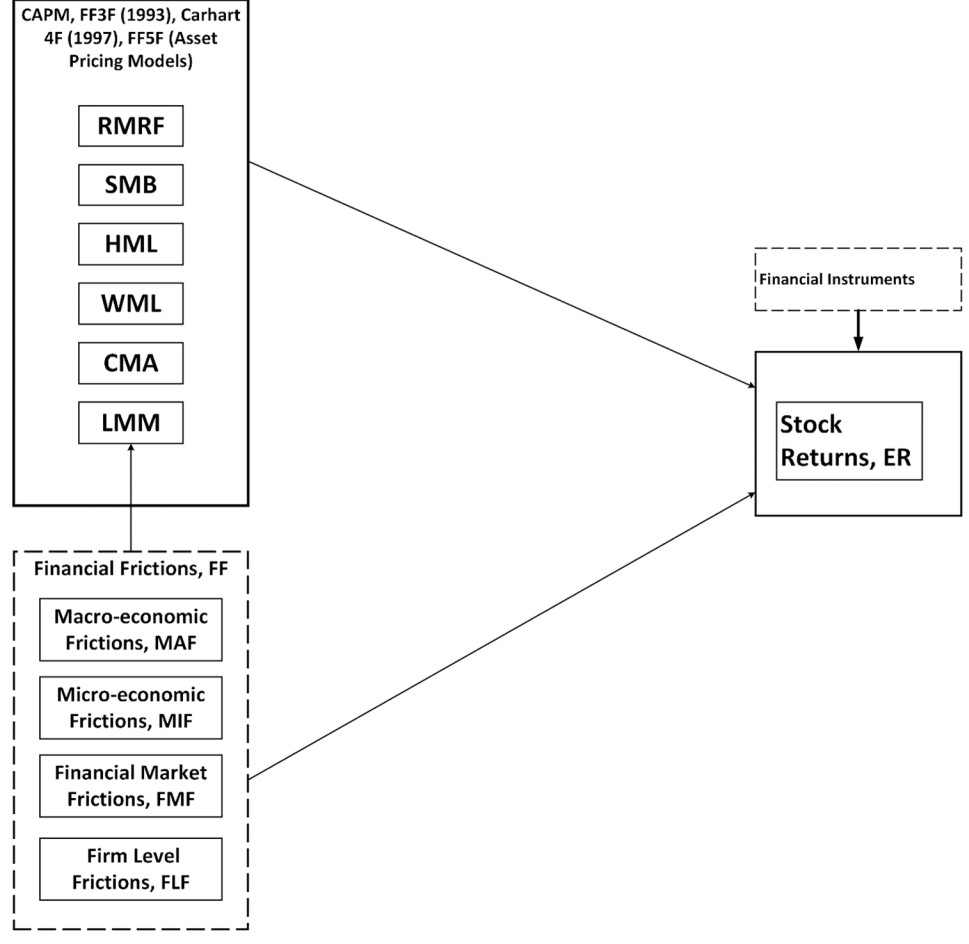

**Fig 1. Theoretical framework.**

financial market, and firm level frictions. Stock returns in every country are compared and explained under the financial instruments of the countries like treasury bill risk free rate and financial institutions lending interest rate.

### 3.2. Hypothesis development

The following hypotheses have been proposed in the light of research questions, study objectives, and theoretical frameworks.

> $H_0$: *There is no effect of various levels of financial frictions on the stock returns of manufacturing firms in South Asian countries when incorporating these frictions in different asset pricing models.*

## 4. Methodology

### 4.1. Study sample

This study investigates the impact of financial frictions on non-financial sector firms in emerging South Asian economies (China, India, Pakistan, Bangladesh, and Sri Lanka). The focus on the non-financial sector stems from its higher sensitivity

to these frictions, unlike the financial sector which acts as a conduit for these frictions. Emerging economies, with their amplified exposure to market imperfections, particularly financial frictions, and considerable growth potential, necessitate this examination. These selected South Asian nations face unique challenges, like underdeveloped financial markets with limited access to finance (especially for manufacturing firms, hindering investment and growth), weaker legal and regulatory frameworks exacerbating information asymmetry and transaction costs, and a substantial informal sector facing even greater financial constraints [50]. The varying explanatory power of the financial friction factor across countries can be attributed to differences in institutional and financial development as it is observed in case of China and India which are higher financially developed and are less prone to these frictions as firms have greater access to diverse funding sources [51]. However, Pakistan, Bangladesh and Sri Lanka are less financially developed and have more financial frictions. Countries with more mature legal systems and higher investor protections tend to show lower financial friction effects on stock returns. Moreover, country-specific characteristics such as institutional quality, corporate governance, creditor rights, market competitiveness, and macroeconomic factors that influence these effects. For instance, better corporate governance and institutional quality tend to reduce financial transaction costs and the small-firm premium, affecting the explanatory power of financial frictions on stock returns differently by countries by their diverse macroeconomic factors. Furthermore, these economies' vulnerability to external shocks is amplified by financial frictions, underscoring the importance of studying these frictions to promote economic growth by improving access to finance and reducing capital costs. Such research can also inform policies promoting financial inclusion and mitigating inequality, as financial frictions disproportionately affect smaller firms and marginalized groups. Data is collected from 735 manufacturing companies listed on local stock exchanges (310 from China, 200 from India, 100 from Pakistan, 50 from Bangladesh, and 75 from Sri Lanka) with the largest market capitalization, spanning the period 2008−2024. Data collection excluded in 2019−2020 to avoid the impact of the COVID-19 pandemic as most of the firms shut down their operations. As Covid-19 period is characterized by unprecedented levels of volatility, significant market disruptions, and potentially irrational investor behavior directly linked to the unfolding COVID-19 pandemic [52]. Including this period, with its outlier characteristics, could potentially skew our results and obscure the underlying long-term relationships we aim to investigate. By excluding these exceptional years, we aim to provide a more robust and stable analysis of the typical market dynamics. The study acknowledges significant cross-country economic and firm-level heterogeneity, causing individual country analyses.

## 4.2. Data sources

The Refinitiv Data Stream is the source of data on the firms included in the sample. The same data are also cross verified from the company's financial statement, which is accessible on the firm's website, as well as from the databases of the nation's security and exchange commission; additionally, any missing data are added.

Data on a country's macroeconomic, microeconomic, and financial markets are gathered from the World Bank and IMF databases. The official databases of the countries, such as central banks, security and exchange commission websites, and other statistical bureau databases, is also used to cross-verify macroeconomic and microeconomic data points. Table 1 tells that how each variable is operationalized in this study in the context of previous studies.

## 4.3. Measurement of financial frictions

**4.3.1. Macroeconomic frictions.** While prior research has proxied macroeconomic frictions using variables such as oil prices [58], exchange rate risk [57], tariffs and transportation costs [59], the mortgage debt-to-income ratio [60], the cost of debt, and price and wage stickiness [61], however, the loan-to-net asset ratio appears to be the superior indicator for capital-intensive manufacturing firms in emerging Asian economies. Consequently, the total debt-to-net assets ratio is adopted here as a proxy for macroeconomic frictions, reflecting the significance of debt as a primary capital requirement relative to a firm's net assets. This choice is further justified by the inclusion of both small and large firms in the sample, precluding the separate consideration of total debt.

**Table 1. Measure of variables used in this study.**

| Variables | Measures | Ref. |
|---|---|---|
| Macroeconomic Frictions, MAF | $MAF = \dfrac{Total\ Loan\ (Short\ term\ and\ Long\ term)}{Total\ Assets}$ | [53] |
| Microeconomic Frictions, MIF | $MIF = \dfrac{\sum(Trade\ Credits,\ accounts\ payables,\ finance\ cost\ )}{Total\ Assets}$ | [54] |
| Financial Market Frictions, FMF | $FMF = TED\ Spread$ | [55] |
| Firm Level Frictions, FLF | $FLF = \dfrac{\sum(labor,\ production,\ operating,\ and\ selling\ costs)}{Total\ Assets}$ | [56] |
| Expected Stock Returns, $R_{it}$ | $R_{it} = \dfrac{(MR_{it_2} - MR_{it_1})}{MR_{it_1}}$ | [2–4] |
| Small minus big, SMB | $SMB = \left[\left(\frac{S}{L} - \frac{B}{L}\right) + \left(\frac{S}{N} - \frac{B}{N}\right) + \left(\frac{S}{H} - \frac{B}{H}\right)\right]/3$ | [2,3] |
| High minus low, HML | $HML = \left[\left(\frac{S}{H} - \frac{S}{L}\right) + \left(\frac{S}{H} - \frac{S}{L}\right)\right]/2$ | [2–4] |
| Winner minus looser, WML | $WML = \left[\left(\frac{S}{W} - \frac{S}{L}\right) + \left(\frac{B}{W} - \frac{B}{L}\right)\right]/2$ | [3,4] |
| Conservative minus aggressive, CMA | $CMA = \left[(SC + BC) - (SA + BA)\right]/2$ | |
| Least minus more, LLM | $LMM = \left(\left(\frac{LF_{1i} + LF_{2i} + LF_{3i} + LF_{4i} + LF_{5i}}{5}\right) - \left((MF_{1i} + MF_{2i} + MF_{3i} + MF_{4i} + MF_{5i})/5\right)\right)$ | [22,56,57] |

Note: TED spread is the spread between 3-month LIBOR and the 3-month Treasury bill rate. $R_{it}$ is the excess stock return of $i$ firm at time $t$, $MR_{it_2}$ is the monthly return of firm at current month and $MR_{it_1}$ is the monthly return of the same firm at previous month. SMB is the excess return on the portfolio of small caps firms less the big caps firm. HML is the excess return on higher valued firms minus the lower valued firms. WML is the excess return on the portfolio of firms which were winning in the last period less looser firms. CMA is the excess return of firms with conservative investment less the firms with aggressive investment strategies. S/L is the portfolio of small firms with lower value, B/L is the portfolio of big firms with lower value, S/N is the portfolio if small firms with neutral value, S/H is the portfolio of small firms with higher value. Similarly, S/W is the portfolio of small firms with winning returns, S/L is portfolio of small firms with looser return, B/W is the portfolio of big firms with winners return and B/L is the portfolio of bigger firms with looser return. SC is the portfolio of small firms with conservative investment return and BC is the portfolio of big firms with conservative investment firms. SA is portfolio of the small firms with aggressive investment and BA is portfolio of big firms with aggressive investment strategies. Whereas, S is from small firm, B is for big firm, H is for higher valued firm, L for lower value firm, N for neutral, C is for conservative, and A is for aggressive investment firms. LF is for lower frictional firms and MF is more frictional firms.

**4.3.2. Microeconomic frictions.** Prior research has measured microeconomic frictions through indicators like risk-weighted capital requirements, bank holdings of safe securities, and risk-based capital requirements [62], alongside trade credit and collateral constraints [13,63], and the restrictive effects of labor regulation on industry [64,65]. Bachmann and Ma [66] also investigated microeconomic frictions by examining plant-level investment spikes. In contrast, this study quantifies microeconomic frictions in manufacturing firms using a newly developed metric. This ratio is calculated by aggregating short-term trade credits, accounts payable, and financing costs from diverse sources (suppliers, merchandisers, financial institutions, and banks), and then normalizing this sum by the firm's total asset value. The interpretation of this metric is that a higher value reflects reduced financial constraints, facilitating smoother financial operations and better access to funding. Conversely, a lower value suggests substantial financial limitations and difficulties in conducting inter-firm activities. This novel measurement of microeconomic frictions offers a significant contribution to understanding the financial well-being and operational effectiveness of manufacturing companies.

**4.3.3. Financial market frictions.** Various proxies have been employed to estimate financial market frictions, including specific fiscal taxes, real transaction costs for investors, the capital gains tax rate [67–70], and the basis (the spread between credit default swap and corporate bond rates) for a significant number of firms [71]. The Treasury Eurodollar spread (TED Spread), defined as the yearly difference between inter-bank loan rates and government bond yields, has also been utilized as an independent variable to represent the level of financial market frictions Li, Liu [42], and Matvos, Seru [72]. Furthermore, Matvos, Seru [72] measured financial market frictions using the excess bond premium (EBP), the TED Spread, and the yield differential between corporate bonds with Baa and Aaa ratings. Therefore, in this study same measure of TED spread used to estimate financial market friction.

**4.3.4. Firm level frictions.** The presence of firm-level frictions can be ascertained by examining constraints on cash flows, impediments to business operations [73], difficulties in achieving optimal capital adjustments, elevated labor and production costs, agency costs arising from managerial and directorial behavior, and limitations imposed by an untrained and unskilled workforce [44,74]. As an illustration, Neumeyer and Perri [75] and Mendoza and Smith [76] assessed firm-level frictions by focusing on working capital financial constraints, which compel businesses to seek borrowing to finance a portion of their wage expenditures or the prepayment of intermediate inputs. Therefore, in this study total labor, production, operations and selling cost to total assets are proxied for firm level friction.

## 5. Financial models

### 5.1. Time series estimated generalized least squares, EGLS models

According to the CAPM, only market or systematic risk has an impact on investment returns. This is because unsystematic or asset risk can be eliminated through investment diversification [77]. The CAPM showed a linear relationship between risk and return, where the beta is a measure of market risk [78]. However, despite several attempts to develop substitute models based on hypotheses and more realistic models that correct CAPM's shortcomings in principle, CAPM still ranks among the most significant asset models [79]. The following equation gives the measure of stock return through the CAPM.

$$R_{it} - RF_{it} = RF_{it} + \beta_i RMRF_{it} + \mu_{it} \tag{11}$$

where $R_{it}$ is the return on a portfolio in excess of one month risk-free rate, $RF_{it}$ is a risk-free rate, $\beta_i$ is the market beta, and $RMRF_{it}$ is the market risk premium, i.e., average of market return less risk-free rate. These asset models incorporate different factors to better measure the CAPM. For instance, the measurement of the market risk premium is based on the Fama and French [2] three-factor model (FF3F), which adds firms' size and market value to the CAPM's excess return to measure stock returns. According to FF3F, smaller firms or smaller-cap companies outperform larger ones and value stocks perform better than growth stocks. The following equation gives the FF3F model:

$$R_{it} - RF_{it} = RF_{it} + \beta_1 RMRF_{it} + \beta_2 SMB_{it} + \beta_3 HML_{it} + \mu_{it} \tag{12}$$

where SMB is the difference of stock returns of small and large firms and HML is the difference of return of higher valued firms and lower valued firms.

A fourth element, momentum, was added by Mark Carhart [3]. Momentum is the tendency for assets to continue a specific course, whether it is upward or downward. His report suggested that the fourth element improved the accuracy of measuring portfolio returns and was based on research on mutual funds. The equation below gives the Carhart (1997) four-factor model (Carhart 4F).

$$R_{it} - RF_{it} = RF_i + \beta_1 RMRF_{it} + \beta_2 SMB_{it} + \beta_3 HML_{it} + \beta_4 WML_{it} + \mu_{it} \tag{13}$$

Similarly, the Fama and French [4] five-factor model (FF5F) is given as

$$R_{it} - RF_{it} = RF_i + \beta_1 RMRF_{it} + \beta_2 SMB_{it} + \beta_3 HML_{it} + \beta_4 RMW_{it} + \beta_5 CMA_{it} + \mu_{it} \tag{14}$$

Where $RMW_{it}$ is the difference between the diversified portfolio of stocks with robust and weak profitability and $CMA_{it}$ is the conservative minus aggressive investment portfolio return and $\beta_5$ is the coefficient for the $CMA_{it}$ factor.

This study incorporates the $LMM_{it}$ factor in the CAPM, FF3F, Carhart4F, and FF5F asset pricing models. Asset pricing models with $LMM_{it}$ financial frictions factor is given as follows.

$$R_{it} - RF_{it} = RF_i + \beta_1 RMRF_{it} + \beta_2 LMM_{it} + \mu_{it} \tag{15}$$

$$R_{it} - RF_{it} = RF_i + \beta_1 RMRF_{it} + \beta_2 SMB_{it} + \beta_3 HML_{it} + \beta_4 LMM_{it} + \mu_{it} \tag{16}$$

$$R_{it} - RF_{it} = RF_i + \beta_1 RMRF_{it} + \beta_2 SMB_{it} + \beta_3 HML_{it} + \beta_4 WML_{it} + \beta_5 LMM_{it} + \mu_{it} \tag{17}$$

$$R_{it} - RF_{it} = RF_i + \beta_1 RMRF_{it} + \beta_2 SMB_{it} + \beta_3 HML_{it} + \beta_4 WML_{it} + \beta_5 CMA_{it} + \beta_6 LMM_{it} + \mu_{it} \tag{18}$$

A $LMM_{it}$ factor is integrated in these asset pricing factors to determine their efficiency within the framework of financial friction and to estimate whether the models with $LMM_{it}$ outperformed the simple assets pricing model in explaining stock returns, particularly for portfolios sorted on profitability and investment or not. Here, $LMM_{it}$ is the portfolio returns of least minus more frictional asset $i$ at time $t$. $LMM_{it}$ includes macro-, micro, financial market and firm-level frictions. The five-factor model outperformed the three-factor model in explaining stock returns, particularly for portfolios sorted on profitability and investment.

## 5.2. Cross-section fixed effect

Traditional time series regression analysis used in CAPM, Fama and French [2,4] and Carhart [3] is enhanced by incorporating fixed effect and the random effect models. The primary advantage of using fixed effect is its ability to effectively control all time-invariant unobserved factors that might influence the dependent variable. This is crucial because these factors can confound the relationship between the variables. If the unobserved factors are constant over time, fixed effects models produce consistent estimates even if these factors are correlated with the independent variables. Fixed effects focus on explaining the variations within an entity over time, making it useful for understanding how changes in independent variables affect the dependent variable for a specific individual, firm, or country. Fixed effects (FE) models provide a powerful tool to address this challenge [80].

The inclusion of fixed effect estimators effectively removes the group-specific and time-invariant components from the independent variables of financial frictions, i.e., macroeconomic, microeconomic, financial market and firm level frictions by mitigating the issue of multicollinearity. This allows the model to focus on within-group and within-period variation, leading to more efficient and unbiased estimates of the relationships between the financial frictions and firm fundamentals. However, it is important to acknowledge that including a large number of fixed effects can lead to a reduction in degrees of freedom, potentially affecting the precision of the estimates [81]. This trade-off between controlling heterogeneity and statistical efficiency is a key consideration when employing fixed effects models in panel data analysis.

Fixed effect models accommodate heterogeneity in panel data by allowing the intercept to vary across groups or time periods while assuming that the relationship between the independent and dependent variables remains constant within these groups or periods [82]. Therefore, the fixed effect model in the case of asset pricing models are given as follows

$$R_{it} - RF_{it} = \alpha_{1i} + \beta_1 (RMRF)_{it} + \beta_2 LMM_{it} + \varepsilon_{it} \tag{19}$$

$$R_{it} - RF_{it} = \alpha_{1i} + \beta_1 RMRF_{it} + \beta_2 SMB_{it} + \beta_3 HML_{it} + \beta_4 LMM_{it} + \varepsilon_{it} \tag{20}$$

$$R_{it} - RF_{it} = \alpha_{1i} + \beta_1 RMRF_{it} + \beta_2 SMB_{it} + \beta_3 HML_{it} + \beta_4 WML_{it} + \beta_5 LMM_{it} + \varepsilon_{it} \tag{21}$$

$$R_{it} - RF_{it} = \alpha_{1i} + \beta_1 RMRF_{it} + \beta_2 SMB_{it} + \beta_3 HML_{it} + \beta_4 WML_{it} + \beta_5 CMA_{it} + \beta_6 LMM_{it} + \varepsilon_{it} \tag{22}$$

The above-fixed effect models deviate from the traditional pooled regression approach by incorporating a subscript i on the intercept term $\alpha_{1i}$. This subscript denotes a critical distinction—the intercept is no longer assumed to be constant

across all observations. Instead, it is allowed to vary for each cross-sectional firms or time period represented by the index i. This flexibility in the intercept structure enables the model to capture unobserved heterogeneity, a defining characteristic of fixed effects models [81]. By accounting for group-specific or time-varying effects that may influence the dependent variable but are not explicitly included in the model, fixed effect models provide a more nuanced understanding of the relationships being estimated variables [83].

### 5.3. Cross-section random effect models

Compared with the fixed effects model, the random effects model offers an alternative approach for capturing unobserved heterogeneity. Fixed effects treat the intercept as unique for each group (cross-section) or time period, requiring a large number of dummy variables [84].

The random effects model, however, views these group-specific intercepts as random variations around a common average effect. It incorporates this random variation into the error term of the model rather than using separate dummy variables [85]. This approach leads to a more streamlined model with fewer parameters, avoiding the potential issue of reduced degrees of freedom observed in fixed effects models with many dummy variables.

Random effects acknowledges that our lack of knowledge about specific group effects can be addressed statistically through the error term rather than through individual intercepts for each group.

Unlike fixed effects models that treat the intercept $\alpha_{1i}$ as unique for each group (cross-section) or time period, the random effects model takes a different approach. It assumes that these group-specific intercepts $\alpha_{1i}$ are not truly fixed but rather variations around a common average effect $\alpha_1$. In simpler terms, the random effects model sees the $\alpha_{1i}$ in each group as the average effect $\alpha_{1i}$ plus some random "noise" specific to that group.

Therefore, $\alpha_{1i}$ in this case is as follows

$$\alpha_{1i} = \alpha_1 + v_i$$

Now, putting the above value of $\alpha_{1i}$, we obtain the following random effect equations of asset pricing models.

$$R_{it} - RF_{it} = \alpha_1 + v_i + \beta_1(RMRF)_{it} + \varepsilon_{it} \tag{23}$$

$$R_{it} - RF_{it} = \alpha_1 + v_i + \beta_1 RMRF_{it} + \beta_2 SMB_{it} + \beta_3 HML_{it} + \varepsilon_{it} \tag{24}$$

$$R_{it} - RF_{it} = \alpha_1 + v_i + \beta_1 RMRF_{it} + \beta_2 SMB_{it} + \beta_3 HML_{it} + \beta_4 WML_{it} + \beta_5 LMM_{it} + \varepsilon_{it} \tag{25}$$

$$R_{it} - RF_{it} = \alpha_1 + v_i + \beta_1 RMRF_{it} + \beta_2 SMB_{it} + \beta_3 HML_{it} + \beta_4 WML_{it} + \beta_5 LMM_{it} + \beta_6 LMM_{it} + \varepsilon_{it} \tag{26}$$

In the random effects model, $v_i$ represents the random component of the intercept that varies across individuals but is constant over time. It captures unobserved individual-specific effects and is assumed to be uncorrelated with the independent variables. The inclusion of $v_i$ in the model allows for more efficient estimation of the factor loadings, provided the assumptions of the random effect, RE model are satisfied.

The choice between fixed and random effects depends on the specific research question and the nature of the data. The Hausman test is often used to decide which model is more appropriate. The Hausman test helps to determine if the unobserved effects are correlated with the independent variables. If the test rejects the null hypothesis, it suggests that fixed effects are more appropriate. If not, random effects might be preferred. Panel data, which tracks multiple entities (like firms, countries) over time, is a treasure trove of information. However, it comes with a challenge: unobserved heterogeneity. This refers to unmeasured factors that can vary across entities (e.g., differences in financial indicators across firms) or time periods [86]. Ignoring this heterogeneity can lead to biased and inconsistent estimates in regression analysis.

## 5.4. Pilot study

A pilot study is performed on the manufacturing firm data of China, Inda and Pakistan by using the same linear models as used by Fama and French [2], Carhart [3], and Fama and French [4] asset pricing models. The results of pilot study are not represented here due to redundancy however their graphs are presented here to depict the trend of stock return in these assets pricing models. The Fig 2, Fig 3, and Fig 4 shows the expected return on the manufacturing firm of China, India and Pakistan. In this analysis CAPM, Fama and French three factor (FF3F) model, Carhart four factor model (Carhart 4F), and Fama and French five factor (FF5F) models. Comparison is presented with and without least minus more, LMM frictional factor. The illustrations show that these models highlight significant inclination and overestimation of asset pricing models without LMM factor as compared to the asset pricing models with LMM. Moreover, these models also follow the same pattern with and without LMM, i.e., when LMM Factor is not incorporated these asset pricing models show higher return. However, FF5F gives little diverse outcome. This pilot study by using traditional asset pricing models allowed us to further incorporate least minus more, LMM factor in the existing models. However, the precise impact of LMM factor is measured and refined in the main study models with fixed and random effect models.

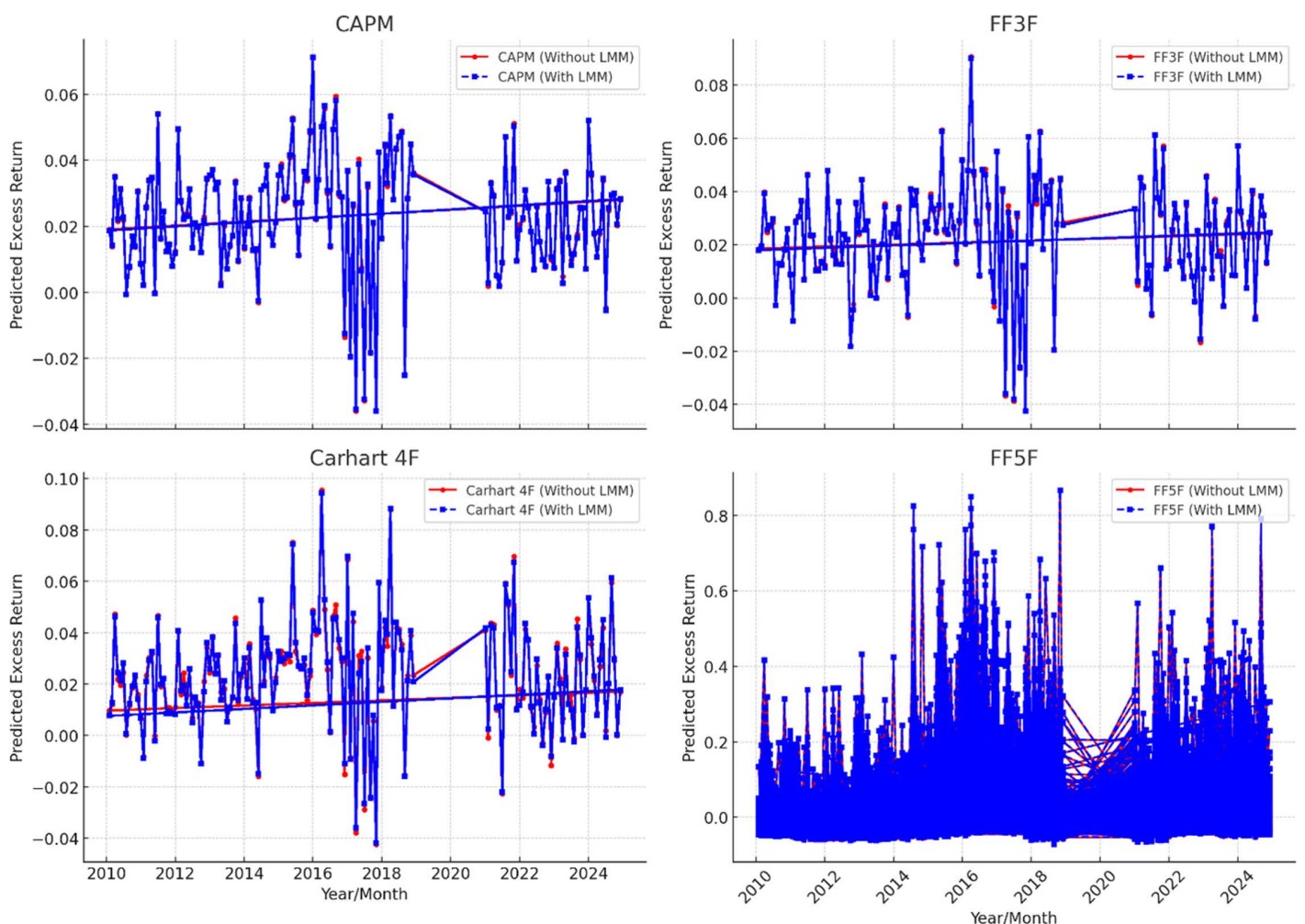

**Fig 2. Asset Pricing Models without and with LMM factor (China).**

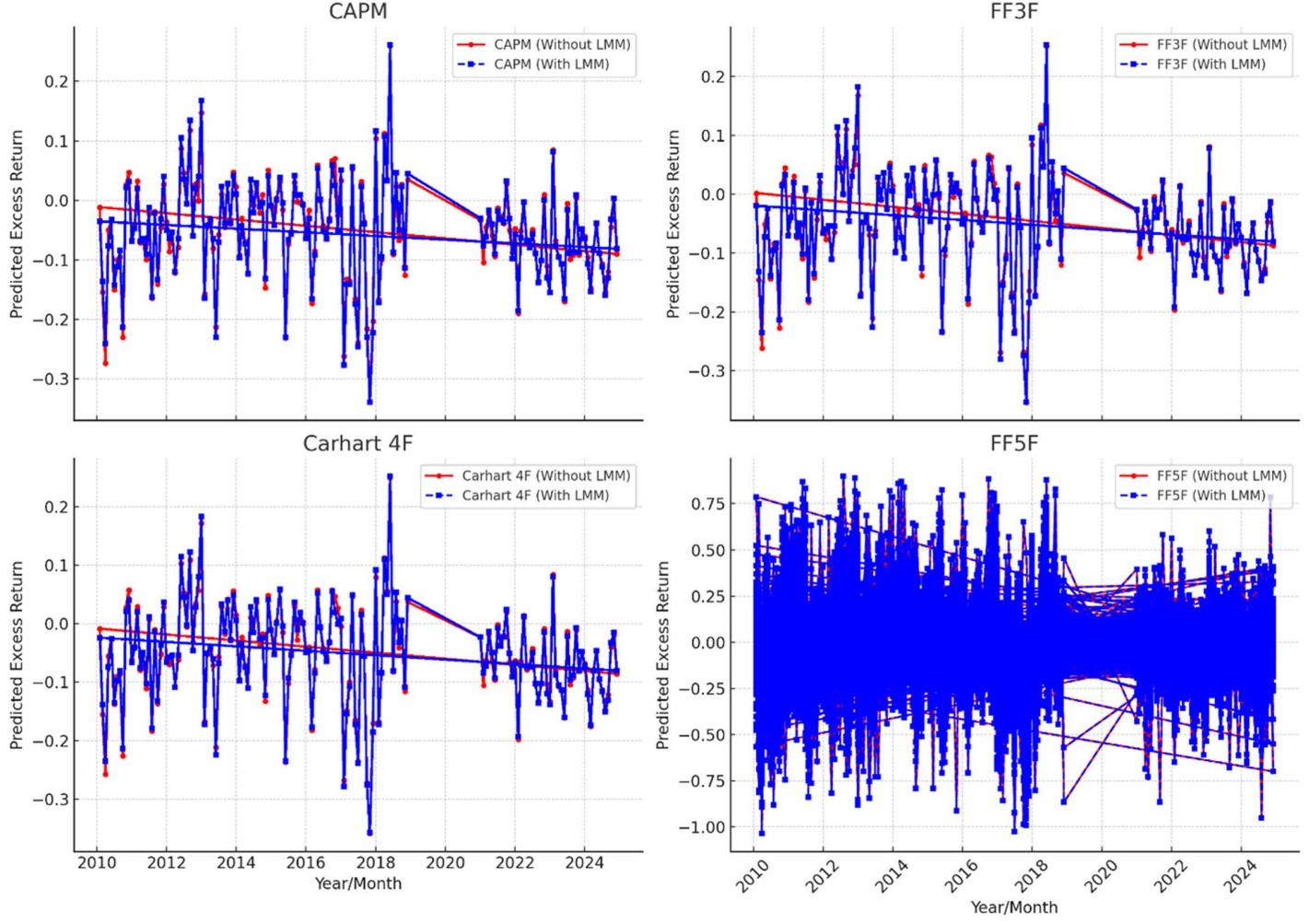

**Fig 3. Asset pricing models without and with LMM factor (India).**

## 6. Diagnostic test

Hausman test, redundant fixed effect test, Pesaran test, Wooldridge test and Wald coefficient restriction test are applied in this study to investigate the efficiency and robustness of the applied models and significance of the results. To test autocorrelation Wooldridge test is applied. Durbin Watson test is also applied on the data to check autocorrelation. Data skewness, and kurtosis are also checked to identify the outliers and data trend. The Hausman test serves as a corner-stone in panel data analysis, providing a statistical framework for choosing between fixed and random effects models that address unobserved heterogeneity (group-specific effects) in different ways [87]. The Hausman test hinges on this crucial assumption of errors in the random effects model. The null hypothesis ($H_0$) states that the random effects model is valid (no correlation in errors). The alternative hypothesis ($H_1$) suggests a violation of the assumption, implying potential bias in the random effects estimator.

The redundant fixed effects test helps to determine whether fixed effects are absolutely necessary or not. In panel data analysis, where data is collected for the same firms over time, a crucial decision involves choosing the right model. One possibility is the pooled error-corrected least squares (EGLS) model, which assumes a constant intercept (average

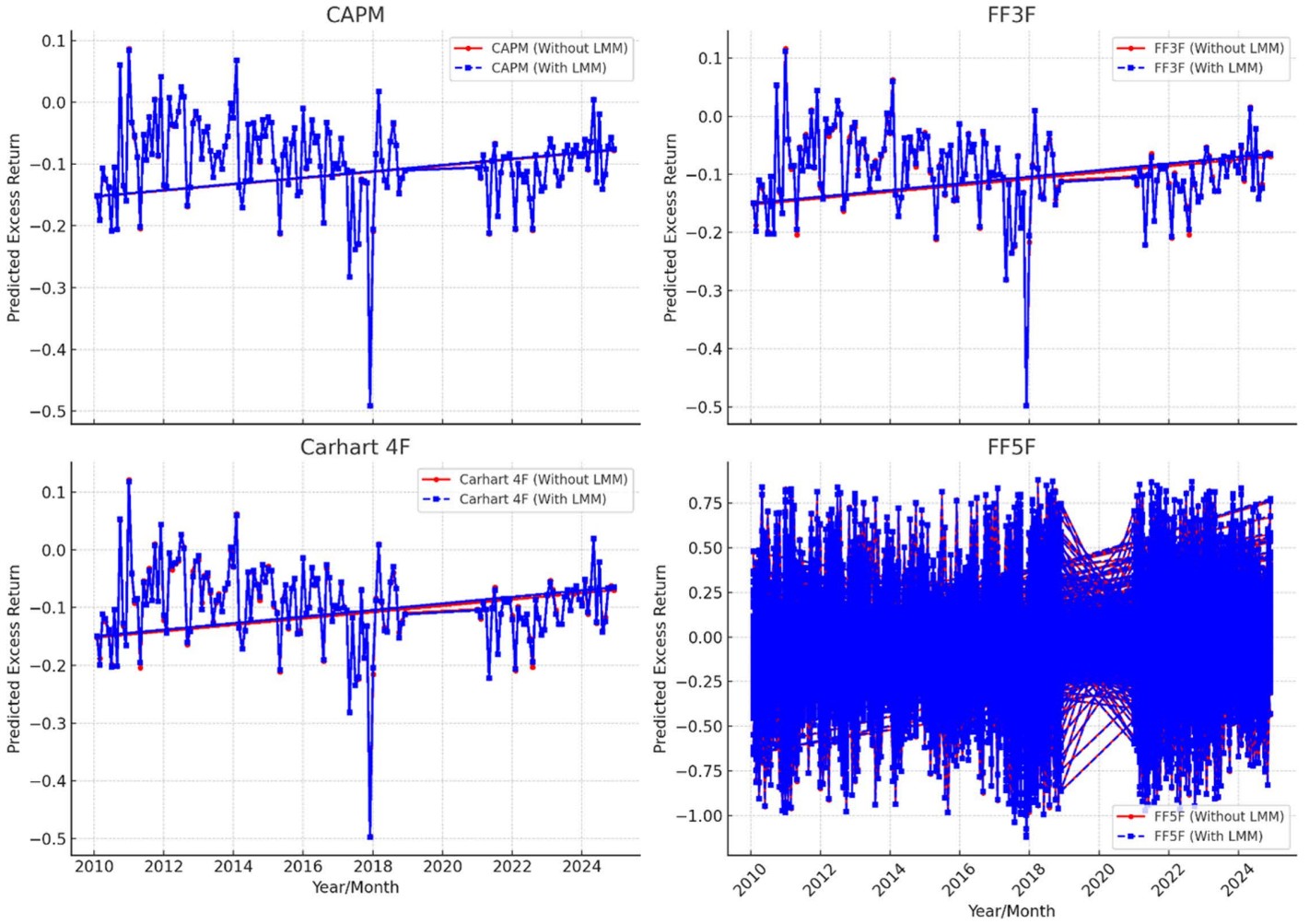

**Fig 4. Asset Pricing models without and with LMM factor (Pakistan).**

effect) across all units. Another option is the fixed effects model, which allows the intercept to vary for each unit, capturing group-specific effects.

The Wald coefficient restriction test measures the discrepancy between the unrestricted estimates and the restrictions imposed by the null hypothesis. A higher value shows a larger discrepancy between the unrestricted estimates and the restrictions imposed. The Pesaran test is a valuable tool in panel data econometrics for detecting cross-sectional correlation among panel units. The test helps identify whether there is cross-sectional dependence in the errors of panel-data models or not. Pesaran test is applied on the data to test the Cross-section correlation among the study variables. However, for cross-section correlation of error terms Hausman Test is applied. The approach of Wooldridge uses the residual first differences in regression. The initial differentiation of the data in the model eliminates the effects at the individual level, covariate duration, and consistency. The Wooldridge test, developed by Semykina and Wooldridge [83], tackles a key challenge in estimating dynamic panel data models, the presence of unobserved heterogeneity (individual-specific effects of auto-correlation). These effects, if not addressed, can lead to biased and inconsistent estimates.

# 7. Results and analysis

The initial step involved the evaluation of descriptive data, followed by estimation of stock return through CAPM, FF3F, Carhart 4F and FF5F asset pricing models. Asset pricing models are estimated through traditional estimated generalized least square models and their extensions with fixed effect and random effect models. Details of the estimated models and their diagnostic tests are shown below.

## 7.1. Descriptive statistics

Descriptive statistics provide an analysis of the characteristics and organization of the data of individual variables. The descriptive statistics are used to assess the fundamental features of all the data, including the data stationarity, multicollinearity, mean, standard deviation, skewness, and level of dispersion through the kurtosis of the variables, as presented in Table 2. Due to redundancy and unnecessary features of pretests like data stationarity and multicollinearity results are not reported here. However, to investigate the suitability for further empirical test these tests are performed initially. The variance inflation factor (VIF) test is applied to test the multicollinearity of the study variables [88]. Results of VIF test show that there is no multicollinearity among the variables. Since for certain variables data is in logarithmic form which reduces the chances of multicollinearity. Moreover, in this panel data analysis this study used fixed effect and random effect models which can sometimes help to mitigate the negative impact of multicollinearity [89].

**Table 2. Descriptive statistics.**

| Var. | China | | | | India | | | | Pakistan | | | |
|---|---|---|---|---|---|---|---|---|---|---|---|---|
| | Mean | Std.Dev | Skew. | Kurtosis | Mean | Std.Dev | Skew. | Kurtosis | Mean | Std.Dev | Skew. | Kurtosis |
| MAF | 0.121 | 0.190 | 2.458 | 6.012 | 0.358 | 0.259 | 0.808 | −0.398 | 0.257 | 0.217 | 0.707 | −0.312 |
| MIF | 0.121 | 0.190 | 2.458 | 6.012 | 0.358 | 0.259 | 0.808 | −0.398 | 0.257 | 0.217 | 0.707 | −0.312 |
| FMF | 0.927 | 0.651 | 0.702 | −1.030 | 3.902 | 2.080 | 0.253 | −1.172 | 0.008 | 0.006 | 1.960 | 3.420 |
| FLF | 0.736 | 3.266 | 0.814 | −0.946 | 0.902 | 2.080 | 0.253 | −1.172 | 0.874 | 1.123 | −0.618 | 0.168 |
| ER | 0.077 | 0.080 | 2.454 | 10.113 | 0.013 | 0.080 | 0.232 | 3.452 | 0.013 | 0.188 | 0.197 | 4.423 |
| RMRF | 0.000 | 0.087 | −0.546 | 1.373 | 0.014 | 0.087 | 0.241 | 3.434 | 0.014 | 0.081 | −1.262 | 6.564 |
| SMB | −0.015 | 0.079 | −0.205 | 2.466 | 0.065 | 0.079 | 1.955 | 2.453 | 0.065 | 0.341 | −0.798 | 20.375 |
| HML | −0.002 | 0.074 | −0.097 | 3.087 | 0.006 | 0.074 | 0.150 | 1.253 | 0.006 | 0.230 | −0.004 | 10.133 |
| WML | 0.145 | 0.133 | 0.811 | 1.128 | 0.007 | 0.133 | 0.658 | 3.008 | 0.007 | 0.519 | 4.218 | 22.305 |
| LMM | −0.010 | 0.052 | 0.634 | 0.683 | 0.219 | 0.052 | 1.306 | 3.531 | 0.219 | 0.150 | −0.007 | 0.846 |
| Var. | Bangladesh | | | | Sri Lanka | | | | | | | |
| | Mean | Std.Dev | Skew. | Kurtosis | Mean | Std.Dev | Skew. | Kurtosis | | | | |
| MAF | 0.327 | 0.244 | 0.670 | −0.698 | 0.351 | 0.251 | 0.753 | −0.529 | | | | |
| MIF | 0.198 | 0.233 | 1.457 | 1.250 | 0.408 | 0.854 | −0.188 | −0.545 | | | | |
| FMF | 0.335 | 0.527 | 0.001 | −0.702 | 0.656 | 2.584 | −3.057 | 8.642 | | | | |
| FLF | 0.482 | 0.742 | −0.314 | −0.647 | 0.384 | 0.219 | −0.742 | 4.384 | | | | |
| ER | 0.018 | 0.185 | 0.384 | 2.343 | 0.011 | 0.153 | 0.724 | 4.033 | | | | |
| RMRF | 0.006 | 0.064 | 0.509 | −0.532 | 0.012 | 0.076 | 0.361 | −0.189 | | | | |
| SMB | 0.001 | 0.075 | −0.364 | 1.109 | −0.013 | 0.092 | −0.166 | 0.021 | | | | |
| HML | −0.011 | 0.068 | 0.543 | 1.538 | −0.019 | 0.137 | −0.512 | 2.417 | | | | |
| WML | 0.169 | 0.171 | 1.517 | 2.793 | 0.190 | 0.170 | 0.490 | 1.964 | | | | |
| LMM | 0.010 | 0.100 | 0.454 | 1.079 | 0.011 | 0.068 | 0.025 | 0.141 | | | | |

Note: Table 1 gives the descriptive statistics of the data. MAF is macroeconomic frictions, MIF is microeconomic frictions, FMF is Financial Market frictions, FLF is Firm level frictions, ER is expected stock return, RMRF is market return less risk free rate, SMB is small minus big factor, HML is high minus low factor, WML is winner minus looser factor, and LMM is least minus high frictions factor.

The analysis reveals that certain variables, such as macro- and microeconomic frictions, and size, show considerable positive standard deviations. This shows a high level of variability in the data, resulting in a decrease in kurtosis and a reduction in the mean dispersion of the curve. The skewness statistics indicate that the data deviates toward one side. The observed positive skewness in the dataset can be attributed to the presence of cross-sectional variations and heterogeneity within the panel data. In China and Pakistan firm size (FS) variables shows slightly positive skewness, which is an indication that the mean is greater than the median. This is because the mean is affected by the outliers (the higher values) and it is pulled towards the right. To remove such outliers, data is further cross verified for any such unrealistic values against different data sources such outliers are removed.

In Pakistan SMB and WML show highly positive kurtosis values. This positive kurtosis shows a distribution flatter tailed than normal, except for higher residuals. This higher kurtosis is because of panel data heterogeneity. To address this heterogeneity fixed effect and random effect models are introduced in this study.

## 7.2. Asset pricing factors with and without LMM

The diagnostic tests conducted for the empirical analysis reveal distinct outcomes across the studied countries, i.e., China, India, Pakistan, Bangladesh, and Sri Lanka, based on the application of traditional estimated generalized least squares, fixed effects, and random effects models. In the cases of China, India, and Pakistan, the fixed effects model provides the most statistically significant and robust specification for estimating the asset pricing models. This decision is confirmed by the Hausman test, which systematically compares the fixed effects and random effects models as shown in Table 7. The test results yield a statistically significant probability value, thereby rejecting the null hypothesis in favor of the alternative hypothesis, which supports for the use of the fixed effects model. The superiority of the fixed effects model in these countries suggests the presence of unobserved heterogeneity that is time-invariant and correlated with the explanatory variables, making fixed effects a more robust estimation technique which is better than pooled time series and cross-section approach adopted in the Fama and French [2,4] three and five factor models.

Conversely, in Bangladesh, the diagnostic tests indicate that the results across all three applied models are highly insignificant. This lack of significance is primarily due to the presence of autocorrelation and multicollinearity issues, which undermine the reliability and efficiency of the estimates. Autocorrelation violates the assumption of independence in error terms, while multicollinearity among explanatory variables inflates standard errors, leading to imprecise coefficient estimates. These econometric challenges necessitate further diagnostic and remedial measures, such as the inclusion of lagged variables or the application of robust standard errors, to improve model performance.

In the case of Sri Lanka, the random effects model is found to provide more statistically significant estimates compared to the fixed effects and EGLS models. This suggests that the unobserved heterogeneity in Sri Lanka is uncorrelated with the explanatory variables, making random effects a more suitable choice.

The panel estimation results of the CAPM in Table 3 shows that for all the countries that display the results of cross-sectional fixed and random effect models. In China the alpha value significantly reduced in the securities return by 23.81% in CAPM, 17.59% in FF3F, 3.7% in Carhart 4F, and 13.63% in the FF5F models with LMM as shown in Table 3, Table 4, Table 5 and Table 6. Alpha value is insignificant in India in FF3F however, Pakistan and Sri Lanka alpha value is also significantly reduced in the asset pricing models. This indicates that a part of the excess returns previously accounted for conventional asset pricing models remains unexplained in the absence of the LMM factor. This implies that these frictions constitute a non-trivial and statistically significant component in the accurate estimation and modeling of asset pricing dynamics.

## 7.3. Alpha, Market (RMRF), Size (SMB), and Value Factor (HML)

In China when LMM factor is introduced in the CAPM model alpha value is reduced by 19.2%, 5% it is reduced in India, 4.3% it is reduced in Pakistan, and 4.7% it is reduced in Bangladesh. Alpha values are significant in all the models and countries except for Bangladesh. In FF3F model alpha value is reduced by 4% in China, while it remains same in India and Sri Lanka and it reduced by 23.8% in Pakistan. Alpha values show the same results in Carhart4F, FF5F models with

**Table 3. CAPM Model without and with least minus more (LMM) frictional factor.**

| Variables | China Without LMM | China With LMM | India Without LMM | India With LMM | Pakistan Without LMM | Pakistan With LMM | Bangladesh Without LMM | Bangladesh With LMM | Sri Lanka Without LMM | Sri Lanka With LMM |
|---|---|---|---|---|---|---|---|---|---|---|
| **alpha** | 0.026 | 0.021 | 0.002 | 0.001 | 0.023 | 0.022 | 2.32E-6 | 1.18E-6 | 0.021 | 0.020 |
|  | (56.91)** | (67.56)** | (18.45)* | (3.63)** | (32.81)** | (21.85)* | (1.72) | (0.39) | (12.23) | (11.27) |
| **RMRF** | 0.055 | 0.052 | 0.091 | 0.092 | 0.181 | 0.160 | 1.000 | 1.000 | 0.014 | 0.014 |
|  | (79.29)** | (77.39)** | (17.52)** | (14.18)** | (10.04)** | (98.82)** | −0.67 | −1.06 | (0.11)* | (66.67)** |
| **LMM** | – | −0.013 | – | −0.023 | – | −0.0493 | – | −0.024 | – | −0.031 |
|  |  | (77.39)** |  | (14.18)** |  | (98.82)** |  | (1.06) |  | (66.67)** |
| **Adj R-sq** | 0.423 | 0.561 | 0.356 | 0.482 | 0.282 | 0.375 | 0.111 | 0.126 | 0.344 | 0.387 |
| **S.E** | 0.132 | 0.132 | 0.141 | 0.140 | 0.179 | 0.180 | 0.078 | 0.084 | 0.135 | 0.134 |
| **Derbin Watson** | 2.756 | 3.001 | 2.069 | 2.086 | 2.203 | 2.349 | 1.754 | 0.947 | 2.139 | 2.16 |
| **F Value** | 22.82** | 601.6** | 86.35** | 196.64** | 324.16** | 104.36** | 120.33** | 119.45** | 31.95** | 448.32** |

Note. **p<0.01, *p<0.05. Relevant t values are reported in parenthesis. Alpha is the single factor alpha. RMRF is a market risk premium, which is equal to average market return minus risk free factor. LMM is the least minus frictional asset pricing factor

**Table 4. Fama and French three factor model with and without least minus more frictional factor (LMM).**

| Var. | China Without LMM | China With LMM | India Without LMM | India With LMM | Pakistan Without LMM | Pakistan With LMM | Bangladesh Without LMM | Bangladesh With LMM | Sri Lanka Without LMM | Sri Lanka With LMM |
|---|---|---|---|---|---|---|---|---|---|---|
| **alpha** | 0.025 | 0.024 | 0.001 | 0.001 | 0.021 | 0.016 | 0.000 | 0.000 | 0.011 | 0.011 |
|  | (31.14)** | (29.35)** | (2.50) | (6.57) | (98.67)** | (3.71)** | (7.64) | (7.75) | (1.38) | (1.25) |
| **RMRF** | 0.051 | 0.050 | 0.018 | 0.013 | 0.021 | 0.018 | 1.000 | 1.000 | 0.015 | 0.017 |
|  | (71.71)** | (70.06)** | (146.16)** | (124.84)** | (619)** | (100.18)** | (3.17) | (1.75) | (67.83)** | (59.41)** |
| **SMB** | 0.038 | 0.032 | 0.011 | 0.011 | 0.044 | 0.042 | 0.000 | 0.000 | 0.125 | 0.121 |
|  | (49.26)** | (49.20)** | (16.43)** | (24.25)** | (3.67)** | (19.96)** | (0.00) | (1.47) | (9.38)** | (12.93)** |
| **HML** | 0.017 | 0.013 | 0.007 | 0.007 | 0.075 | 0.066 | 0.000 | 0.000 | 0.079 | 0.059 |
|  | (14.41)** | (15.66)** | (1.61)** | (1.19) | (10.34)** | (16.95)** | (0.00) | (2.71) | (2.03) | (5.14) |
| **LMM** | – | −0.026 | – | −0.037 | – | 0.314 | – | −0.008 | – | −0.041 |
|  |  | (15.66)** |  | (7.19)* |  | (16.95)** |  | (2.71) |  | (5.14) |
| **Adj R-sq** | 0.479 | 0.491 | 0.357 | 0.480 | 0.383 | 0.488 | 0.111 | 0.126 | 0.348 | 0.397 |
| **S.E** | 0.128 | 0.181 | 0.139 | 0.122 | 0.108 | 0.102 | 0.791 | 0.721 | 0.113 | 0.111 |
| **Derbin Watson** | 2.032 | 2.030 | 2.073 | 2.092 | 2.193 | 2.319 | 2.076 | 2.076 | 2.152 | 2.176 |
| **F Value** | 341.62** | 316.06** | 86.52** | 686.93** | 118.69** | 328.98** | 179.33** | 310.35** | 32.13** | 173.38** |

Note. **p<0.01, *p<0.05. Relevant t values are reported in parenthesis. Var. is variable. alpha is Fama and French three factor alpha. RMRF is a market risk premium, which is equal to average market return, RM minus risk free factor. SMB is small minus big factor which is equal to return on portfolio of small minus big firms, HML high minus low factor which is equal to return on portfolio of high value firms minus low value firms. LMM is least minus more frictional factor.

and without LMM factor. While In China the market excess return, RMRF, is reduced by 5.7% in CAPM, 2.0% in FF3F, 4.8% in FF5F while there is no change seen in Carhart 4F model. However, market factor increased in India by 1.1% while it reduced by 11.6% in Pakistan and remain same in Sri Lanka in the CAPM model. In China the market factor, RMRF, is reduced by 5.7% in CAPM, 2.0% in FF3F, 4.8% in FF5F while there is no change seen in Carhart 4F model. However, market factor increased in India by 1.1% while it reduced by 11.6% in Pakistan and remain same in Sri Lanka in the CAPM model.

**Table 5. Carhart Four Factor Model with and without Least minus more frictional factor (LMM).**

| Variables | China Without LMM | China With LMM | India Without LMM | India With LMM | Pakistan Without LMM | Pakistan With LMM | Bangladesh Without LMM | Bangladesh With LMM | Sri Lanka Without LMM | Sri Lanka With LMM |
|---|---|---|---|---|---|---|---|---|---|---|
| Const. | 0.025 | 0.021 | 0.019 | 0.019 | 0.103 | 0.103 | 0.000 | 0.000 | 0.013 | 0.013 |
| | (18.66)** | (18.94)** | (5.29)** | (6.98)** | (24.48)** | (19.23)** | (14.98) | (12.70)** | (7.82)** | (6.31)** |
| RMRF | 0.049 | 0.049 | 0.087 | 0.088 | 0.144 | 0.142 | 1.000 | 1.000 | 0.011 | 0.010 |
| | (71.33)** | (71.49)** | (143.22)** | (122.15)** | (89.77)** | (59.88)** | (0.23) | (344.39)** | (65.79)** | (57.28)** |
| SMB | 0.0375 | 0.036 | 0.013 | 0.012 | 0.034 | 0.020 | 0.000 | 0.000 | 0.117 | 0.115 |
| | (47.06)** | (47.16)** | (14.13)** | (22.18)** | (1.07) | (4.82)** | (1.44) | (19.80)** | (12.69)** | (15.78)** |
| HML | 0.010 | 0.012 | 0.006 | 0.006 | 0.041 | 0.044 | 0.000 | 0.000 | 0.045 | 0.042 |
| | (14.38)** | (14.41)** | (1.23) | (1.51) | (0.97) | (0.57) | (2.14) | (21.86)** | (4.65)** | (7.27)** |
| WML | 0.025 | 0.025 | 0.014 | 0.017 | 0.064 | 0.062 | 0.000 | 0.000 | 0.040 | 0.041 |
| | (5.46)** | (5.48)** | (5.10)** | (5.07)** | (8.88)** | (10.18)** | (0.59) | (8.94)** | (7.420) | (5.56)** |
| LMM | 0.036 | 0.040 | 0.138 | 0.235 | 0.0213 | 0.026 | 0.000 | 0.000 | 0.019 | 0.022 |
| | (3.01)** | (3.01)** | (17.92)** | (14.02)** | (22.29) | (23.23)** | (0.69) | (0.00) | (11.69)** | (11.09)** |
| Adj R-sq | 0.495 | 0.561 | 0.443 | 0.572 | 0.547 | 0.576 | 0.111 | 0.126 | 0.357 | 0.425 |
| S.E | 0.118 | 0.118 | 0.138 | 0.139 | 0.258 | 0.258 | 0.756 | 0.865 | 0.108 | 0.113 |
| Derbin Watson | 2.032 | 2.030 | 2.077 | 2.094 | 2.358 | 2.467 | 2.180 | 1.926 | 2.161 | 2.185 |

Note. **p<0.01, *p<0.05. Relevant t values are reported in parenthesis. RMRF is market risk premium, which is equal to average market return, RM minus risk free factor. SMB is return on portfolio of small minus big firms, HML is return on portfolio of High minus low firm. WML is the return on the portfolio of winner minus loser factor. LMM is the return on portfolio of least minus more frictional firms.

**Table 6. Fama and French five factor model with and without least minus more frictional factor (LMM).**

| Variables | China Without LMM | China With LMM | India Without LMM | India With LMM | Pakistan Without LMM | Pakistan With LMM | Bangladesh Without LMM | Bangladesh With LMM | Sri Lanka Without LMM | Sri Lanka With LMM |
|---|---|---|---|---|---|---|---|---|---|---|
| Const. | 0.025 | 0.022 | 0.007 | 0.007 | 0.103 | 0.103 | 0.000 | 0.000 | 0.013 | 0.013 |
| | (24.56)** | (37.52)** | (34.52)** | (19.34)** | (63.92)** | (81.28)** | (1.02) | (2.70) | (32.82)** | (21.31)* |
| RMRF | 0.051 | 0.049 | 0.087 | 0.088 | 0.144 | 0.142 | 1.000 | 1.000 | 0.011 | 0.010 |
| | (37.15)** | (64.11)** | (23.39)** | (42.41)** | (92.62)** | (52.19)** | (0.11) | (4.36)* | (92.79)** | (93.63)* |
| SMB | 0.037 | 0.030 | 0.013 | 0.011 | 0.034 | 0.024 | 0.000 | 0.000 | 0.132 | 0.110 |
| | (62.27)** | (57.13)** | (93.59)** | (78.20)** | (1.02) | (71.92)** | (1.06) | (3.19)* | (4.82)* | (83.38)** |
| HML | 0.014 | 0.012 | 0.006 | 0.004 | 0.041 | 0.038 | 0.000 | 0.000 | 0.042 | 0.033 |
| | (26.11)** | (27.38)** | (0.22) | (0.51) | (0.03) | (0.28) | (2.02) | (0.36) | (83.65)** | (77.37)** |
| WML | 0.022 | 0.020 | 0.016 | 0.011 | 0.062 | 0.057 | 0.000 | 0.000 | 0.040 | 0.040 |
| | (65.26)** | (82.45)** | (49.42)** | (83.83)** | (93.42)** | (82.11)** | (0.021) | (0.24) | (4.410)* | (35.16)* |
| CMA | 0.036 | 0.032 | 0.331 | 0.312 | 0.021 | 0.018 | 0.000 | 0.000 | 0.019 | 0.013 |
| | (33.06)** | (82.42)** | (83.19)** | (15.62)** | (82.12)** | (69.82)** | (0.42) | (0.00) | (3.39)* | (21.01)* |
| LMM | – | −0.064 | – | −0.031 | – | −0.062 | – | 0.000 | – | −0.032 |
| | | (62.48)** | | (19.53)** | | (92.21)** | | (0.000) | | (23.21)** |
| Adj R-sq | 0.532 | 0.634 | 0.461 | 0.529 | 0.428 | 0.581 | 0.111 | 0.126 | 0.425 | 0.471 |
| S.E | 0.132 | 0.122 | 0.432 | 0.135 | 0.214 | 0.232 | 0.821 | 0.891 | 0.308 | 0.536 |
| Derbin Watson | 2.068 | 2.044 | 2.055 | 2.006 | 2.021 | 2.298 | 2.001 | 1.926 | 2.332 | 2.005 |

Note. **p<0.01, *p<0.05. Relevant t values are reported in parenthesis. RMRF is market risk premium, which is equal to average market return, RM minus risk free factor. SMB is return on portfolio of small minus big firms, HML is return on portfolio of High minus low firm. WML is the return on the portfolio of winner minus loser factor. CMA is the return on portfolio of conservative minus aggressive factor. LMM is the return on portfolio of least minus more frictional firms.

**Table 7. Diagnostic tests sub model I, II and III.**

| Variables | | China | | India | | Pakistan | | Bangladesh | | Sri Lanka | |
|---|---|---|---|---|---|---|---|---|---|---|---|
| | Test | Chi-Sq. | Prob. | Chi-Sq. | Prob. | Chi-Sq. | Prob. | Chi-Sq. | Prob. | Chi-Sq. | Prob. |
| CAPM | Hausman Test | 2.28E03** | 0.000 | 4111E05** | 0.000 | 5.84E2** | 0.000 | 1.157 | 0.720 | 0.004** | 0.000 |
| | Redundant Fixed Effect | 61.373** | 0.000 | 0.891* | 0.043 | 69.88** | 0.000 | 33.238 | 0.382 | 0.000 | 1.000 |
| CAPM (LMM) | Hausman Test | 3.71E04** | 0.000 | 1.89E06** | 0.000 | 6.19E24** | 0.000 | 1.002 | 0.332 | 0.000** | 0.000 |
| | Redundant Fixed Effect | 213.449** | 0.000 | 0.742* | | 38.445** | 0.000 | 0.000 | 1.000 | 0.011** | 0.000 |
| FF3F | Hausman Test | 7.18E06** | 0.000 | 2.29E11** | 0.000 | 1.61E14** | 0.000 | 8.118 | 0.628 | 0.000* | 0.026 |
| | Redundant Fixed Effect | 43.394** | 0.000 | 0.850 | 0.097 | 90.660 | 0.445 | 0.000 | 1.000 | 0.054** | 0.000 |
| FF3F (LMM) | Hausman Test | 3.01E04** | 1.000 | 4,719E16** | 0.000 | 38.98E4** | 0.000 | 93.119 | 0.281 | 0.000** | 0.000 |
| | Redundant Fixed Effect | | 0.000 | | 0.000 | | | 1.162 | 1.000 | 0.054** | |
| Carhart 4F | Hausman Test | 6.28E05** | 0.000 | 2.28E14** | 0.000 | 33.17E6* | 0.012 | 1.597 | 0.000 | 0.000* | 0.063 |
| | Redundant Fixed Effect | 111.394* | 0.042 | 0.0852* | 0.0478 | 11.534** | 0.000 | 29.991 | 1.000 | 0.145** | 0.000 |
| Carhart 4F (LMM) | Hausman Test | 3.24E03** | 0.000 | 3.32E18** | 0.000 | 2.28E14** | 0.000 | 78.291 | 0.922 | 0.000 | 1.000 |
| | Redundant Fixed Effect | 65.345 | 0.877 | | 0.000 | 51.281 | 0.72 | 314.216 | 0.382 | 0.054** | 0.000 |
| FF5F | Hausman Test | 3.30E03 | 0.055 | 5.92E24** | 0.000 | 2.28E14** | 0.000 | 2.009 | 0.765 | 0.757** | 0.004 |
| | Redundant Fixed Effect | 82.394** | 0.000 | 0.895* | 0.0497 | 0.325** | 0.000 | 56.757 | 1.000 | 0.000 | 1.000 |
| FF5F (LMM) | Hausman Test | 8.29E04* | 0.037 | 2.14E14** | 0.000 | 2.28E14** | 0.000 | 76.221 | 0.262 | 0.072* | 0.013 |
| | Redundant Fixed Effect | 38.619** | 0.000 | | 0.000 | | 0.000 | 0.226 | 0.004 | 0.054** | 0.002 |

Note. **$p<0.01$, *$p<0.05$. Chi-Sq. is a chi-square estimate of Hausman and Redundant fixed effect test. CAPM is a capital asset pricing model, and CAPM (LMM) is capital asset pricing model with least minus more frictional factor. FF3F is Fama and French three factor model, and FF3F (LMM) is Fama and French three factor model with least minus more frictions. Carhart 4F is Carhart four factor model, and Carhart 4F (LMM) is Carhart four factor model with least minus more frictional factor. FF5F is Fama and French five factor model whereas FF5F (LMM) is Fama and French five factor model with least minus more frictional factor. In this table diagnostic test are run through Hausman Random effect model and Redundant fixed effect models to estimate the dominance of fixed and random effect in the asset pricing models.

Similarly, in the manufacturing firms of China, while introducing two more components, SMB and HML, in FF3F and factor loading of RMRF decreases by 5.1%. In the Carhart 4F model, this value decreases by 5.0%, while the same value decreases by 4.9% when we introduce the least minus more financial frictions factor (LMM) in FF5F model, indicating the overestimation of excess market returns on all the securities. Small minus big, SMB fund returns are positively related to stock returns, and their values are 3.9% and 3.8% in the FF3F and Carhart 4F models, respectively, while their value with the LMM in the five-factor model decreases to 3.8%.

Similarly, In China the values of HML in FF3F and Carhart 4F are 1.7% and 1.1%, respectively, which shows that HML is positively associated with stock returns. The value of HML decreases to 0.1% in five factor models with LMM which shows that the value of HML is also overestimated in the FF3f and Carhart 4F models. In India the HML value is 0.7% and 0.8% in FF3F and Carhart 4F models respectively. The value of HML in five factor models is 0.6%. HML value in Pakistan is 7.5% and 6.4% FF3F and Carhart 4F model respectively while it is 4.1% in five factor model with LMM. The value of HML is not significant in Bangladesh while it is 5.9%, 5.5% and 4.3% in FF3F, Carhart 4F and five factor model respectively. The value of HML is also significantly decreased in the five-factor model with LMM factor which tells that these asset pricing models overestimate the stock returns while with the induction of LMM factor this anomaly is significantly removed.

## 7.4. Momentum factor, WML

The value of WML is positively associated with stock returns, and its value China in the Carhart 4F model is 2.8%, whereas it decreases to 2.5% in the five-factor model with the LMM. The value of WML in India is 2.5% and 1.4% in

Carhart 4F and five factor model with LMM. In Pakistan the value of WML is 7.3% and 6.4% in Carhart 4F and five factor model with LMM. The value of WML in Sri Lanka is 5.1% and 4.1% in Carhart 4F and five factor model with LMM. This shows that the LMM factor significantly reduced the value of WML in Carhart 4F model in all the countries and depicts that Carhart 4F model overestimates the value of WML in emerging countries.

### 7.5. Least minus more frictional factor, LMM

The results show that with the integration of LMM factor in asset pricing model, it significantly reduced the CMA factor estimates, showing that our estimation in the models. In China CMA is reduced by 11.11%, in India it is reduced by 5.74%, in Pakistan it is reduced by 14.29%, whereas in Sri Lanka it is reduced by 31.58%. This shows that LMM has a negative impact on CMA factors, and it significantly reduced its explanatory power due to financial friction. These results suggest that the CMA factor is more influential in certain markets (like China and India) and less so in others (like Bangladesh), and the inclusion of financial frictions (LMM) seems to strengthen the significance of the CMA factor in most cases, especially in larger emerging markets.

The LMM factor is negatively associated with the individual stock returns. The value of LMM in China is −3.6%, in India it is −13.8, in Pakistan it is −23.6, in Bangladesh its value is not significant while in Sri Lanka its value is −2.2%. The LMM adjusts the overestimated values of the four factors by reducing them in the overall model of the factor model by −4.0%. This standard error estimate in all the models is reduced from 0.132 to 0.118 in the CAPM to and it reduces to 0.128 in both the FF3F and Carhart 4F models. A slightly positive autocorrelation is observed in the 2.02 model. The overall results of all the models are significant, as indicated by F statistics.

In these countries the variation in the stock return due to CAPM is explained by 42.3% in China, 35.6% in India, 28.2% in Pakistan, the CAPM result of Bangladesh are not significant, and 34.4% in Sri Lanka. While in US in stock return the variation in explained by CAPM model is about 70% as mentioned by Fama and French [90]. They also mentioned that the CAPM has been a useful tool for understanding risk and return, its empirical performance is not perfect, explaining only around 70% of the variation in stock returns. This high explanatory power of CAPM in US is due to free market while the lower explanator power of South Asian economies is due to control and in efficient market capital markets. Moreover, these economies are also closed economies [91]. In India the variation in stock return due to market factor in CAPM is recorded as 35.6%, it is 28.2% in Pakistan, and 34.4% in Sri Lanka. Similarly the variation in stock return due to explanatory variables FF3F is recorded as 47.9% in China, 35.7% in India, 38.2% in Pakistan and 34.8 in Sri Lanka. Variation in stock return due to explanatory factors Carhart 5F is recorded as 38.0% in China, in India it is recorded as 42.4%, in Pakistan it is recorded as 44.0% while it is 42.8% in Sri Lanka. Similarly, variation in stock return due to explanatory variables in five factor model with LMM is recorded as 49.5% in China, 44.3% in India, 54.7% in Pakistan and 42.5% in Sri Lanka. These results show that the variation in the stock return is better explained by the models with the least minus more frictional factors.

### 7.3. Diagnostics tests

In the case of India, cross section redundant fixed effects are more significant according to the diagnostic tests performed. The Hausman test of redundant effects revealed no significant effects, and this supported the alternative hypothesis in favor of redundant fixed effects.

Most of the results for Bangladesh are insignificant in all the models and in the diagnostic test. All the values of the weighted least squares, fixed effect, and random effect model is not significantly different, as indicated by the T values. In Bangladesh, asset pricing factors have a significant value of 100%, which is observed for excess market returns. The SMB, HML and WML factors do not work in the case of Bangladesh. The value of the LMM is also zero in this country. This table shows that the Durbin–Watson values of all the models ensure the highest positive autocorrelation in the data.

According to the diagnostic test results in Table 8 show that the cross-sectional redundant fixed effect model results are significant in the CAPM, FF3F, and Carhart 4F models, while the cross-sectional random effect results model is significant in the five-factor model with the LMM. Therefore, only five-factor model results are taken from cross-sectional random effect models. The standard error also moderate in all the models, and there is a slightly positive autocorrelation observed in all the orders from Durbin Watson statistics. All the models are significant and robust, as indicated by the F-statistics.

This shows that the estimated coefficients included in the five-factor model with the LMM depict the true factor loadings of all the factors, i.e., the RMRF, SMB, HML, WML, and LMM. The above analysis concludes that the market risk premium and RMRF coefficients explain the average required rate of return in the market. The SMB beta shows that small stocks outperform large cap stocks in the market. The HML beta values show that growth stocks possess lower returns than value stocks. The WML beta depicts whether the market is illiquid and facing a volatile bearish trend. The LMM factor beta shows that stocks with lower financial frictions in terms of either firms' internal or external factors have higher returns, and vice versa.

The results of Wald coefficient restrictions in Table 8 shows that LMM factor used in the asset pricing models are significant as neither beta coefficient is zero nor it is equal to one except for Sri Lanka. The test shows that dropping LMM factor from the asset pricing model significantly affect the factor loadings of asset pricing model. It is also showed that all the values of chi-square are more significant with improved t statistics and less standard error by using the LMM factor in the asset pricing models as compared to the asset pricing models without LMM factor.

## 8. Conclusion

In Pakistan, companies are below their optimal efficiency level, and they have house opportunities and room to expand their production. According to the CAPM, the market risk premiums of the Fama and French three factor model and Carhart four factor models are highest in Pakistan, which is also due to the highest inflation and interest rates. In Sri Lanka, larger firms generate greater stock returns than do larger firms in other countries. Companies with higher book-to-market ratios (value companies) are receiving greater stock returns in China than in other economies. These HML factor results

**Table 8. Wald coefficient restriction test.**

| Var. | China | | | India | | | Pakistan | | | Bangladesh | | | Sri Lanka | | |
|---|---|---|---|---|---|---|---|---|---|---|---|---|---|---|---|
| | Chi-Sq. Values | Std. Err. | t value | Chi-Sq. values | Std. Err. | t value | Chi-Sq. values | Std. Err. | t value | Chi-Sq. values | Std. Err. | t value | Chi-Sq. value | Std. Err. | t value |
| C(1)=0 | 0.018 | 0.006 | −18.66* | 0.003 | 0.001 | 3.71** | 0.031 | 0.001 | −24.47** | 0.000 | 0.000 | 14.99 | 0.013 | 0.002 | 7.82* |
| C(2)=0 | 0.500 | 0.007 | 71.36** | 0.994 | 0.007 | 143.14** | 0.824 | 0.009 | 89.77** | 1.000 | 0.000 | 4.07E + 17 | 0.999 | 0.015 | 65.79** |
| C(3)=0 | 0.375 | 0.008 | 47.06** | 0.062 | 0.004 | 14.15** | 0.003 | 0.003 | −1.07 | 0.000 | 0.000 | 30.45 | 0.176 | 0.014 | 12.69** |
| C(4)=0 | 0.117 | 0.008 | 14.38** | 0.006 | 0.005 | 1.22 | 0.004 | 0.004 | 0.97 | 0.000 | 0.000 | 24.17 | 0.045 | 0.010 | −4.65** |
| C(5)=0 | 0.025 | 0.007 | 5.46** | 0.014 | 0.003 | 5.10** | 0.014 | 0.002 | 8.87* | 0.000 | 0.000 | −9.62 | 0.048 | 0.006 | −0.42 |
| C(6)=0 | 0.032 | 0.012 | 5.46** | 0.014 | 0.003 | 5.10** | 0.014 | 0.002 | 8.87* | 0.000 | 0.000 | −9.62 | 0.048 | 0.006 | −0.42 |
| C(1)=1 | 1.018 | 0.001 | 159.24* | 0.997 | 0.001 | 119.32** | 1.031 | 0.001 | 82.83* | 1.000 | 0.000 | −1.78E + 18 | 0.987 | 0.002 | −59.22** |
| C(2)=1 | 0.501 | 0.007 | 71.48** | 0.006 | 0.007 | 0.86** | 0.176 | 0.009 | 19.13** | 0.000 | 0.000 | −45.12 | 0.001 | 0.015 | −0.09 |
| C(3)=1 | 0.625 | 0.008 | 78.53** | 0.938 | 0.004 | 21.20** | 1.003 | 0.003 | 34.89* | 1.000 | 0.000 | −1.76E + 17 | 0.824 | 0.014 | −59.37** |
| C(4)=1 | 1.117 | 0.008 | 37.71* | 0.994 | 0.005 | 26.41** | 0.996 | 0.004 | 23.87** | 1.000 | 0.000 | −1.64E + 17 | 1.045 | 0.010 | 107.58* |
| C(5)=1 | 1.025 | 0.005 | 22.07* | 1.014 | 0.003 | 381.61* | 0.986 | 0.002 | 63.37** | 1.000 | 0.000 | −4.15E + 17 | 1.048 | 0.006 | 161.79* |
| C(6)=1 | 1.052 | 0.001 | 41.22* | 2.391 | 0.003 | 18.28* | 0.986 | 0.011 | 63.37** | 1.000 | 0.000 | −2.52E + 11 | 0.492 | 0.003 | 11.79* |

Note: Wald coefficient restriction test that is performed to identify the robustness of the coefficient of the five-factor model with least minus more frictions. The coefficients are tested on two hypotheses, $H_0$ and $H_1$. $H_0$ states that all the $\beta$ values are equal to zero, and $H_1$ states that all the $\beta$ values are equal to one. Wald these statistics show that no beta is equal to zero or equal to one except for Bangladesh.

are the same as those presented by Huang and Liu [92]. Similarly, those companies that had the highest return in the last period (WML) also had the maximum return in the current period, which is also the highest in China.

By including frictional factor the alpha value is significantly reduced in all the asset pricing models which suggest that the portfolios are generating less return than what would be expected based on their exposure to the model's factors. It indicates a decline in the manager's ability to generate returns above the benchmark predicted by the models. Alpha is often attributed to a manager's skill in selecting undervalued stocks (stock selection) or in anticipating market movements (market timing). A decreasing alpha could imply a deterioration in these skills or that the strategies employed are no longer as effective in the current market frictional conditions [93]. The market factor in traditional models assumes friction-less trading. Introducing friction can reduce the explanatory power of the market factor because friction distorts arbitrage, preventing efficient pricing and making some assets appear riskier than they are under perfect markets [94]. Investors demand additional compensation for bearing liquidity risk or transaction costs, which is not captured by market beta alone [95].

When frictional factors are introduced, smaller companies, which are often less liquid, could be more heavily impacted by transaction costs, which may diminish their premium or render it less predictable [96]. Therefore, the SMB factor could lose significance in a model that accounts for such costs.

Similarly, when frictional factors are considered, the relative performance of value stocks might be less pronounced, as buying and selling these stocks might incur higher costs. Thus, value stocks might not always outperform as expected, especially when these costs eat into potential returns [97]. Financial frictions can erode the profitability of momentum strategies by increasing trading costs as mentioned by de Roon and Szymanowska (2012), who show that incorporating transaction costs into asset pricing models reduced the predictability of returns from momentum strategies. They found that transaction costs as low as 35 basis points could reconcile the observed return predictability with asset pricing models. Same results are reported by the Endri and Endri (2020) who incorporates transaction costs into the traditional Fama-French three-factor model. Their findings indicate that while the traditional factors (market, size, and value) remained significant, the inclusion of transaction costs provided a more accurate representation of expected returns, especially for stocks with higher trading frictions.

Incorporating the financial frictions as least negative, the LMM factor in the CAPM, FF3F, and Carhart 4F models significantly reduced the values of the returns on the market factors in the FF3F and Carhart 4F models. This indicates that these models overestimate the value of stock returns without economic friction. Therefore, it is concluded that FFs are negatively associated with stock returns and, when they are incorporated into the asset pricing model, provide better estimates of portfolio returns.

In the market, firms are also not optimal performers in the presence of financial friction. All the asset pricing models employed in this study showed higher returns on the stock, but with the induction of the LMM, this higher return is adjusted to a lower value. However, the values indicate that without the presence of financial frictions, asset pricing models show higher returns than larger firms do, but these estimates are also overestimated. The SMB and HML are positively related to stock returns in all the countries. This factor also significantly decreased in the presence of the LMM factor, supporting the fact that without the presence of the LMM, it was overestimated and adjusted to the optimal level in the presence of the LMM factor. The WML factor is positively related to stock returns. It is assured that these factors overestimate in the Carhart 4F model, which adjusts itself to the best value in the induction of the factor model with the LMM. These results are the same as those of Ozdagli [48] who mentioned that financially constrained firms, such as those with limited access to capital or higher borrowing costs, hinder these firms' ability to invest and grow, thereby reducing their stock performance and often resulting in lower average stock returns. These studies considered monetary policy and higher capital and borrowing costs as constraints for higher stock returns. However, this study has mentioned macroeconomic and microeconomic frictions related to these constraints.

The study assumes that financial frictions negatively impacts firm value, stock returns, and capital structure. While this is accepted, the analysis could be strengthened by examining the specific mechanisms through which financial frictions affect these variables. For example, how do financial frictions affect a firm's cost of capital, investment decisions, and access to external financing? A deeper exploration of these mechanisms would provide a more nuanced understanding of the impact of financial frictions [98]. The text extends the assumptions of agency theory to macro- and micro-level firm interactions. While this is an interesting theoretical contribution, the analysis could be strengthened by empirically evaluating these extended assumptions. For example, does the level of government intervention or the strength of stakeholder relationships affect agency costs? Empirical evidence would provide stronger support for the theoretical extension of agency theory. By addressing these assumptions, the analysis could be significantly strengthened, providing a more comprehensive and nuanced understanding of the relationship between financial frictions, firm-specific fundamentals, and the role of financial development and firm strategies in mitigating these frictions.

It was assumed that the agency theory assumptions are not only valid for the firm level and financial market level frictions as agency costs, but these are also prevailing in the macro-economic and micro-economic level as well. These frictional costs at the macro, micro and financial market level are due to conflict of interest between manufacturing firms and the market players which are government bodies like tax authorities, corporate regulatory bodies like security and exchange commission, and trade associations. These authorities and regulatory bodies' interest is to increase the government treasury while the interest of manufacturing companies is to maximize their or shareholders wealth. Therefore, government bodies levied taxes and tariffs on the firms while manufacturing firms keep their prices and profit margins high. As mentioned above the results show that financial frictions has the same negative impact on firm fundamentals as the agency costs have. Theoretically, agency theory assumptions are tested at the macro, micro and financial market level in this study.

Results shows that frictions can distort a firm's optimal debt-equity mix, leading to higher financial risk and lower profitability as mentioned by Hennessy and Whited [26]. Similarly frictions can negatively impact investor sentiment and increase uncertainty, leading to lower stock prices and reduced returns for shareholders as revealed by Alfaro, García-Santana [99]. High inflation erodes the value of money, creating uncertainty about future prices and discouraging lending and investment. This uncertainty can lead to increased information asymmetry through which lenders struggle to assess creditworthiness when future prices are unpredictable. Lenders demand higher interest rates to compensate for inflation risk, making borrowing more expensive. Businesses become hesitant to invest due to uncertain returns, further hindering economic activity. High interest rates often accompany high inflation as central banks try to curb rising prices. However, high rates can also exacerbate financial frictions by increasing borrowing costs and making it making it more difficult for firms and individuals to access credit [100]. Similarly, reducing asset values results in declining collateral values, making it harder to secure loans.

Asset pricing models also do not predict real stock returns, as these models do not account for the different levels of FFs. Therefore, this study incorporates a novel factor of least minus more (LMM), which better predicts stock returns than do previously renowned pricing models.

In the current era, firms are more vulnerable and prone to external constraints and costs to achieve greater profitability. Therefore, the government must frame prudential and monetary policies according to the interests of listed companies; for instance, they must reduce inflation in the economy, keep interest rates low, increase exports, and reduce imports. The government should be stable for at least a decade and have larger achievable plans and policies. The government must reduce the revised taxes and tariffs on manufacturing firms. The government has also kept major commodity prices, especially petroleum prices, stable. The government also keeps the national currency in a stable or growing state. The government supports financial development in the economy by introducing new technology and flourishing new industries by allowing foreign investments. The findings offer valuable guidance for policymakers by understanding the mechanisms through which financial frictions affect firm dynamics. Policymakers can design more effective interventions to mitigate the

negative consequences of financial crises and support economic recovery. Policymakers may also need to consider regulatory reforms or interventions to mitigate financial frictions highlighted in this study. This could involve changes in financial regulation, market structure, or incentives to improve market efficiency. Financial frictions can influence the effectiveness of monetary policy transmission mechanisms. Central banks may need to adjust their policies differently to account for these frictions, potentially affecting interest rates and credit availability.

Governments should focus on reducing inflation and interest rates to minimize financial market friction. High inflation increases uncertainty, erodes the value of money, and increases transaction costs. Lower interest rates will reduce borrowing costs for firms, encouraging investment and improving overall market efficiency. Implement monetary policies aimed at controlling inflation and keeping interest rates stable. This will make financing cheaper and reduce liquidity risks, improving firms' ability to invest and grow.

Simplify tax and tariff structures for manufacturing firms to reduce financial constraints. Financial frictions negatively impact firm performance due to government interventions like taxes and tariffs such as lower corporate taxes, streamline tariffs, and provide tax incentives for firms investing in R&D and technology upgrades. This would enable firms to better manage operational costs and enhance their competitiveness. Introduce regulations aimed at enhancing market transparency, improving financial reporting standards, and encouraging the adoption of modern technologies. This will make the market more efficient and reduce the impact of financial friction on firms' stock returns.

Emerging economies need robust financial systems to better absorb financial shocks and reduce the negative impact of liquidity risk on asset pricing for strengthening financial institutions, promote capital market development, and encouraging foreign investment. This will enhance the depth and liquidity of local markets, enabling firms to access better financing options and reducing the overall cost of capital. Results shows that smaller firms are particularly vulnerable to transaction costs and financial frictions. Providing better access to capital will allow these firms to grow and become more efficient. Therefore, government should develop policies to improve access to external financing, especially for Small and Medium Enterprises, such as low-interest loans, venture capital funds, or government-backed guarantees for business loans. This will help them overcome capital constraints and reduce the impact of liquidity risk on their operations.

Governments and policymakers can help reduce financial frictions by creating more stable macroeconomic environments and reducing regulatory burdens. The findings suggest that policies aimed at lowering inflation and interest rates could help firms manage financial frictions more effectively, leading to improved performance [101].

Governments should implement policies to stabilize macroeconomic conditions, such as controlling inflation and reducing government debt. This can help reduce financial market frictions and create a more stable environment for businesses [102]. Firms should invest in research and development, adopt modern technologies, and improve their operational efficiency to enhance productivity growth.

Policymakers can use these microeconomic insights from this research to design policies aimed at reducing financial frictions and promoting a more efficient allocation of resources within emerging economies. This could involve regulatory reforms, incentives for transparency, and measures to enhance financial market development. In short, the microeconomic implications of the article focus on understanding how financial frictions, earnings management, and productivity growth interact to influence firm behavior, market dynamics, investor behavior, and policy formulation within emerging economies. These insights are crucial for stakeholders such as firms, investors, policymakers, and regulators aiming to navigate and improve economic outcomes in these markets.

The present study is subject to limitations concerning the number of metrics employed to examine macroeconomic and microeconomic frictions, with a specific emphasis on external financing and financial constraints. Furthermore, the study is limited to emerging economies in South Asia. By including emerging economies from other continents, the study could have more generalized results as the study may be limited by the size and representativeness of the sample used. Emerging economies can vary widely in terms of economic structure, regulatory environment, and market dynamics.

The limitations identified highlight several areas for potential improvement and deeper exploration in future research. Firstly, the study's reliance on a limited number of metrics to examine macroeconomic and microeconomic frictions focusing mainly on external financing and financial constraints may overlook other critical dimensions of financial frictions such as information asymmetry, legal enforcement, governance quality, and transaction costs. Future studies could improve the robustness of the analysis by incorporating a broader set of variables or constructing a composite index of financial frictions that include internal financing constraints, corporate governance indicators, and market transparency measures.

Secondly, the geographical scope of the study is restricted to emerging economies in South Asia, which limits the generalizability of the findings. Since emerging markets differ widely in their economic structures, regulatory frameworks, and financial systems, expanding the analysis to include other regions such as Latin America, Eastern Europe, or Sub-Saharan Africa would provide a more comprehensive understanding of how financial frictions operate across various contexts. A cross-regional panel analysis could also uncover whether the observed effects are region-specific or universally applicable.

Another significant limitation stems from data incompleteness and inconsistency, particularly affecting firms in Bangladesh and Sri Lanka. Many companies were excluded due to recent listings or delisting's, which may have introduced selection bias into the sample and reduced the representativeness of the findings. To address this, future research should seek alternative or complementary data sources—such as firm-level surveys, private databases, or public filings—and apply data imputation techniques or bootstrapping methods to mitigate the impact of missing data.

Moreover, the available data is also not complete and consistent, which resulted in the exclusion of many companies, and ultimately smaller samples are constructed for Bangladesh and Sri Lanka. Several firms in these countries are also not listed throughout the study period either due to their recent incorporation or delisting due to noncompliance with their regulatory bodies such as the Securities and Exchange Commission.

The study results introduce a new paradigm for future research in the field of real-world financial frictions models, which may be further studied in developed economies such as the US, European countries, or Asian developed countries such as Japan, Indonesia, and Malaysia Korea. Since this study included mostly manufacturing firms, however, its findings may be generalized to the nonmanufacturing sectors for instance, insurance, financial, and telecommunications sectors. To gain a more complete understanding of the economy, future research should focus on integrating findings from the current study into other macroeconomic frameworks and industries, such as new Keynesian models, service sectors, financial institutions, and public sectors.

The analysis section provides a wealth of findings on the relationship between financial frictions and stock return. However, there's potential to enhance the robustness and comprehensiveness of these claims by considering alternative data interpretations and additional analyses which could be exploring more heterogeneity among the firm-specific characteristics. For instance, the current analysis could be enriched by exploring how the impact of financial frictions varies across firms with different characteristics, such as age, industry, and ownership structure.

Conducting sensitivity analyses by varying the sample period, estimation techniques, and model specifications could help assess the robustness of the results to different methodological choices. Therefore, by incorporating these alternative data interpretations and additional analyses, the study can strengthen the validity of its findings and provide a more nuanced and comprehensive understanding of the complex relationship between financial frictions and firm fundamentals in the presence of earnings management, firm productivity growth, and financial development in selected South Asian economies.

## Supporting information

**S1 Data. All countries data set.**
(XLSX)

## Author contributions

**Conceptualization:** Saifullah Khan.

**Data curation:** Adnan Shoaib, Rehan Aftab, Muhammad Yasir, Muhammad Bilal Saeed.

**Formal analysis:** Adnan Shoaib, Rehan Aftab, Muhammad Yasir, Muhammad Bilal Saeed.

**Investigation:** Saifullah Khan.

**Methodology:** Adnan Shoaib, Rehan Aftab, Muhammad Yasir, Muhammad Bilal Saeed.

**Resources:** Rehan Aftab, Muhammad Yasir, Muhammad Bilal Saeed.

**Software:** Adnan Shoaib, Rehan Aftab, Muhammad Yasir, Muhammad Bilal Saeed.

**Supervision:** Saifullah Khan, Rehan Aftab.

**Validation:** Adnan Shoaib, Rehan Aftab, Muhammad Yasir, Muhammad Bilal Saeed.

**Visualization:** Adnan Shoaib, Rehan Aftab, Muhammad Yasir, Muhammad Bilal Saeed.

**Writing – original draft:** Saifullah Khan.

**Writing – review & editing:** Saifullah Khan.

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
