## [Decision Letter · Decision Letter 0]

Dear Dr. Khan,

Thank you for submitting your manuscript to PLOS ONE. After careful consideration, we feel that it has merit but does not fully meet PLOS ONE’s publication criteria as it currently stands. Therefore, we invite you to submit a revised version of the manuscript that addresses the points raised during the review process.

Please submit your revised manuscript by May 30 2025 11:59PM. If you will need more time than this to complete your revisions, please reply to this message or contact the journal office at plosone@plos.org . A rebuttal letter that responds to each point raised by the academic editor and reviewer(s). You should upload this letter as a separate file labeled 'Response to Reviewers'.A marked-up copy of your manuscript that highlights changes made to the original version. You should upload this as a separate file labeled 'Revised Manuscript with Track Changes'.An unmarked version of your revised paper without tracked changes. You should upload this as a separate file labeled 'Manuscript'.

We look forward to receiving your revised manuscript.

Kind regards,

Ricky Chee Jiun Chia

Academic Editor

PLOS ONE

Journal Requirements:

2. In the online submission form, you indicated that [Complete data set can also be requested from authors upon request].

Reviewers' comments:

Reviewer's Responses to Questions

**Comments to the Author**

1. Is the manuscript technically sound, and do the data support the conclusions?

Reviewer #1: No

Reviewer #2: Yes

Reviewer #3: Yes

Reviewer #4: Yes

2. Has the statistical analysis been performed appropriately and rigorously?

Reviewer #1: No

Reviewer #2: Yes

Reviewer #3: Yes

Reviewer #4: No

3. Have the authors made all data underlying the findings in their manuscript fully available?

Reviewer #1: Yes

Reviewer #2: Yes

Reviewer #3: No

Reviewer #4: Yes

4. Is the manuscript presented in an intelligible fashion and written in standard English?

Reviewer #1: Yes

Reviewer #2: Yes

Reviewer #3: Yes

Reviewer #4: Yes

Reviewer #1: Major comments:

- It is not clear on how the least minus more frictional factor approach is better than traditional approaches, e.g. Bali et al. (2016), to price if the friction factors are priced in stock market. In addition, there is a huge literature that firm characteristics such as size is a proxy for financial frictions, see Ferreira et al. (2023) and reference therein. It is therefore important to properly test your frictional factor with traditional approaches that are more robust than the authors' current approach.

- in regard to the macroeconomic frictions, the authors discuss that macroeconomic policies can either mitigate or amplify the consequences of frictions. On the other hand, Baker et al. (2016) have shown the impact of Economic Policy Uncertainty to economy, and their data is available to China and India. In this line of research, Kundu and Paul (2022) recently report its impact to stock market of G-7 countries. Therefore, the authors may need to consider it whenever available to see if it mitigates or amplify the consequences of these frictions to stock market.

- any reason to only focus on the manufacture firms? the authors should consider all the firms publicly listed in each of 5 countries. if the frictions have the market-wide impact, we should observe similar effects to non-manufacture firms.

- it is not clear why data excludes 2019-2020 period since stock market is still open in that time period.

Minor comments:

- in Equation 11, RMRF should be the expected market risk premium, not average of market returns less risk-free rate. In equation 12 to 26, since your LHS variable is expected value, the RHS shouldn't have the error term.

- not sure how to read Figure 2 to 4, especially the panel on the bottom right of each Figure. The auhors may need to consider another way of data visualization.

Reference

Miguel Ferreira, Timo Haber and Christian Rogiror, Financial constraints and firm size: micro-evidence and aggregate implications, DNB Working Paper No 777/ May 2023

Scott R. Baker, Nicholas Bloom, Steven J. Davis, Measuring Economic Policy Uncertainty, The Quarterly Journal of Economics, Volume 131, Issue 4, November 2016, Pages 1593–1636, https://doi.org/10.1093/qje/qjw024

Srikanta Kundu, Amartya Paul, Effect of economic policy uncertainty on stock market return and volatility under heterogeneous market characteristics, International Review of Economics & Finance, Volume 80, 2022, Pages 597-612,

ISSN 1059-0560, https://doi.org/10.1016/j.iref.2022.02.047.

Turan G. Bali, Stephen J. Brown, Yi Tang, Is economic uncertainty priced in the cross-section of stock returns?,

Journal of Financial Economics, Volume 126, Issue 3, 2017, Pages 471-489, ISSN 0304-405X, https://doi.org/10.1016/j.jfineco.2017.09.005

Reviewer #2: The reviewer believes that this paper has made a significant innovative marginal contribution to the literature by introducing financial friction factors to explain stock returns based on four types of capital asset pricing models (CAMP).

My concerns, comments and suggestions are as follows:

Questions with the definition of variables in Table 1

1 There is no definition of "N" in SMB, and what does S/L refer to?

2 What does Winner Minus Loser (WML) mean, and what does S/W mean?

3. The subscript 1-5 in LMM is ambiguous, does it refer to five types of friction, but the paper only mentions four types of friction generated at the macro, micro, financial market, and corporate level. Is it a combined friction? How to synthesize?

4. How are Small-Big, High-Low, and Conservative-Aggressive defined?

The above variable measures are an important basis for subsequent panel regression.

5. Panel data regression involves the selection of mixed, random effects, and fixed-effect models, and can also consider the heteroskedasticity of disturbed terms, autocorrelation test, and cross-section correlation test. If you use STATA, you can run the following command:

Cross-section correlation tests: xtcsd, fre

Heteroskedasticity test: xttest3

Autocorrelation test: xtserial

If the above problems exist, the panel data regression can be corrected by xtscc.

Reviewer #3: Reviewer #1: The study by Saifullah Khan, Adnan Shoaib, Rehan University Aftab, Muhammad Yasir, Muhammad Bilal Saeed provides a valuable contribution to empirically evaluate the role of various levels of financial friction in explaining stock returns through different asset pricing models. They enhances asset pricing model estimates by incorporating diverse levels of financial friction by introducing a novel least minus more frictional asset pricing factor specifically constructed for emerging economies. But I think the article has the following issues:

Although the article introduces the concept of financial friction, it lacks in-depth analysis of the relationship between different levels of friction. For example, the application of agency theory only stays at the theoretical level and has not been fully validated with empirical data.

2. The data sample is limited to five emerging economies in South Asia, which may limit the generalizability of the research conclusions. Expanding the sample size or increasing the selection of sample data demonstrates the universality of research conclusions.

3.Although a panel data model was used, potential endogeneity issues were not addressed.

4.The results from Bangladesh are mostly not significant, but the study did not explain this difference, weakening the generalizability of the conclusions.

5.The conclusion of the article is relatively vague and does not fully integrate empirical results for in-depth analysis. The policy recommendations are relatively broad, lacking specificity and operability.

6.The article's discussion of research limitations is relatively simple and does not fully consider possible alternative explanations or improvement directions.

Reviewer #4: Review Comments

This manuscript explores the role of financial frictions in explaining stock returns using various asset pricing models in emerging economies. The research topic is both novel and of substantial academic value. However, several areas require further improvement to enhance the overall clarity, coherence, and rigor of the study:

Logic of Empirical Analysis

The analysis of empirical results lacks clarity, particularly in Sections 7.2.1 to 7.2.4. The authors are encouraged to present their findings in a more structured and logical manner. A clearer exposition would significantly improve the readability and interpretability of the results.

Adjusted R-squared Values for Bangladesh

In Tables 3 through 6, the reported Adjusted R-squared values for the CAPM model (with and without the least-minus-more (LMM) frictional factor) for Bangladesh are all shown as 1.0. This appears highly implausible and warrants closer examination. Moreover, the authors should provide a detailed discussion explaining the varying explanatory power of the financial friction factor across different countries. This may include an exploration of country-specific characteristics that influence the effect of financial frictions on stock returns.

Measurement of Financial Frictions

The approach to measuring financial frictions requires further elaboration in both the literature review and methodology sections. A more comprehensive discussion of existing theories and methodologies related to financial frictions is needed, along with a clear justification for the authors' chosen method. This will strengthen the theoretical grounding of the study and reinforce its methodological rigor.

Clarity and Precision of Language

Several portions of the manuscript contain unclear expressions and minor grammatical inaccuracies. The authors should undertake a thorough review of the text to ensure all statements are articulated clearly and are free from language errors. This will improve the overall readability and professionalism of the paper.

Conclusion

In summary, while the study addresses an important and timely topic, it requires substantial revisions to address the issues outlined above.

**Do you want your identity to be public for this peer review?** For information about this choice, including consent withdrawal, please see our Privacy Policy

Reviewer #1: No

Reviewer #2: No

Reviewer #3: No

Reviewer #4: No

---

## [Author Response · Author response to Decision Letter 1]

15 May 2025

All the reviewers comments has been addressed in the "Response to the reviewers" file. and also incorporated in the manuscript file. Thank you

---

## [Editor Report · Decision Letter 1]

Financial frictions and stock return: A novel least minus more frictional factor for asset pricing models in emerging economies

PONE-D-25-10692R1

Dear Dr. Saifullah Khan,

We’re pleased to inform you that your manuscript has been judged scientifically suitable for publication and will be formally accepted for publication once it meets all outstanding technical requirements.

Kind regards,

Ricky Chee Jiun Chia

Academic Editor

PLOS ONE
---

## [Editor Report · Acceptance letter]

PONE-D-25-10692R1

PLOS ONE

Dear Dr. Khan,

I'm pleased to inform you that your manuscript has been deemed suitable for publication in PLOS ONE. Congratulations! Your manuscript is now being handed over to our production team.

Kind regards,

on behalf of

Dr. Ricky Chee Jiun Chia

Academic Editor

PLOS ONE